# On the Generalization of Training-based ChatGPT Detection Methods

## Abstract

ChatGPT is one of the most popular language models which achieve amazing performance on various natural language tasks. Consequently, there is also an urgent need to detect the texts generated ChatGPT from human written. One of the extensively studied methods trains classification models to distinguish both. However, existing studies also demonstrate that the trained models may suffer from distribution shifts (during test), i.e., they are ineffective to predict the generated texts from unseen language tasks or topics. In this work, we aim to have a comprehensive investigation on these methods' generalization behaviors under distribution shift caused by a wide range of factors, including prompts, text lengths, topics, and language tasks. To achieve this goal, we first collect a new dataset with human and ChatGPT texts, and then we conduct extensive studies on the collected dataset. Our studies unveil insightful findings which provide guidance for developing future methodologies or data collection strategies for ChatGPT detection.

## 1 Introduction

ChatGPT (OpenAI) is one of the most popular language models, which demonstrates a great versatility to handle diverse language tasks, including question answering (Tan et al., 2023), creative writing (Bishop, 2023) and personal assistance (Shahriar & Hayawi, 2023). Meanwhile, it also gives rise to an urgent need for detecting ChatGPT generated texts from human written texts to regulate the proper use of ChatGPT. For example, ChatGPT can be misused to accomplish the tasks such as producing fake news or generating fake reviews (Li et al., 2023), leading to public deception. Similarly, ChatGPT can be also used for plagiarism, offending people's intellectual property (Falati, 2023). These misuses of ChatGPT can cause severe negative consequences for our society.

Since the model parameters of ChatGPT are not publicly available, many detection techniques for open-source language models (e.g., DetectGPT (Mitchell et al., 2023), Watermarks (Kirchenbauer et al., 2023)) cannot be utilized for ChatGPT detection. Therefore, a major stream of works (Guo et al., 2023; Chen et al., 2023; Tian et al., 2023b) propose to train classification models on the collected human texts and ChatGPT texts to distinguish each other, which we called "*training-based methods*" in this work. The empirical studies also demonstrate that the trained classifiers can achieve high detection performance under their studied datasets.

However, it is evident from recent works (Yu et al., 2023; Guo et al., 2023) that these training-based methods tend to be overfitted to their training data distribution. For instance, Guo et al. (2023) show that a RoBERTa classification model (Liu et al., 2019) trained on HC-3 dataset (Guo et al., 2023) for detecting ChatGPT answered questions will exhibit a notable accuracy decrease, when it is tested on some specific topics (i.e., finance and medicine). Yu et al. (2023) also find that the detection models trained on HC-3 struggle to detect ChatGPT written news or scientific paper abstracts. In addition, in our work, we also notice that other types of distribution shifts (between training and test distribution) can occur and cause detection performance decrease, which are not identified or adequately discussed in previous works. These distribution shifts include:

- *Prompts to Inquire ChatGPT outputs:* The ChatGPT user can have various prompts to obtain the ChatGPT outputs. For example, when asking ChatGPT to write a movie review, a user can ask "Write a review for the movie <MovieTitle>". Alternatively, they can also let ChatGPT give comments to the movie, via asking ChatGPT to complete a dialogue which reflects the preference of the talkers towards this movie (see Section 3 for more details). The detection models that trained on texts obtained from certain prompts may face texts from other unknown prompts.

- *Length of ChatGPT outputs:* The ChatGPT user can designate and control the length of the output to inquire longer or shorter generated outputs. It is also possible that the (distribution of) lengths of test samples differ from training ones.

In reality, because only a limited number of training data can be collected, the training data cannot fully cover the distribution of test data. Thus, it is critical to deeply understand the detection models' generalization behaviors when the distribution shifts occur. To achieve this goal, we first collect a new text dataset, named *HC-Var (**H**uman **C**hatGPT Texts with **Var**iety)*, which contains human texts and ChatGPT outputs by considering multiple types of variety, including prompts, lengths, topics and language tasks (see Section 3). Facilitated with HC-Var, we can conduct comprehensive analysis on the models' generalization, when facing the aforementioned distribution shifts. Through extensive experiments, we draw key findings and understandings, which provide guidance for developing better methodologies and data collection strategies to assist the success of ChatGPT detection:

- From the pessimistic side, we identify one possible reason that can hurt the detection models' generalization. For the training-based methods, the trained models tend to overfit to some "**irrelevant features**" which are not principal for ChatGPT detection. This overfitting issue can be originated from the "incautious and insufficient" data collection process, which collects ChatGPT texts that are distinct from human texts in these "irrelevant features". In Section 4.3, we conduct a theoretical analysis to deeply understand this phenomenon.
- From an optimistic side, we find the trained models are also capable to extract "**transferable features**", which are shared features that can help detect the ChatGPT generated texts from various topics and language tasks. For example, in Section 5, we show that the models trained on existing topics or language tasks can be leveraged as a source model to accommodate transfer learning (Pan & Yang, 2009; Hendrycks et al., 2019), when it is adapted to unforeseen topics and language tasks.

## 2 RELATED WORKS

In this section, we introduce background knowledge about existing methods for ChatGPT generated text detection, as well as other detection methods for open-source language models. We also discuss existing research findings about the generalization of ChatGPT detection methods.

### 2.1 OPEN-SOURCE LANGUAGE MODEL DETECTION AND CHATGPT DETECTION

For open-source language models such as GPT-2 (Solaiman et al., 2019), and LLaMa (Touvron et al., 2023), since their model parameters are publicly available, information such as model probability scores can be leveraged for detection. For example, DetectGPT (Mitchell et al., 2023) assumes that LLMs always generate the texts with high probability scores. Thus, it manipulates the candidate texts (by editing or paraphrasing) to check whether the model gives a lower probability score. Besides, there are watermarking strategies (Kirchenbauer et al., 2023) which intervene the text generation process to inject watermarks into the generated texts to make them identifiable.

Detecting ChatGPT generated texts is also an important task because of the extraordinary prevalence of social-wide usage of ChatGPT. However, many previously mentioned methods are not applicable due to the lack of access to ChatGPT's model & probability scores. Therefore, plenty of works leverage the ***Training-based Methods*** (Guo et al., 2023; OpenAI, 2019; Chen et al., 2023), to train classification models to predict whether a text $x$ is human-written or ChatGPT generated:

$$\min_f \mathbb{E}\Big[\mathbf{1}(f(x) \neq y)\Big],\ y \sim \{0,1\},\ x \sim \begin{cases} \mathcal{D}_H & \text{if } y = 0 \\ \mathcal{D}_C & \text{if } y = 1 \end{cases} \tag{1}$$

where $\mathcal{D}_H$ and $\mathcal{D}_C$ represent the collected human and ChatGPT texts, respectively. Besides, there are "similarity-based" methods, such as GPT-Pat (Yu et al., 2023) and DNA-GPT (Yang et al., 2023) to compare the similarity of a text $x$ with its ChatGPT re-generated texts. Besides, "score-based methods" such as GPT-Zero (GPTZero.com) and GLTR (Gehrmann et al., 2019) detection ChatGPT texts based on their specific traits. More details of these methods are in Appendix D.

### 2.2 TRAINING-BASED CHATGPT DETECTION UNDER DISTRIBUTION SHIFT

Notably, our work is not the first work studying or identifying the generalization issues of the training-based ChatGPT detection models. For example, the works (Wang et al., 2023; Yang et al., 2023; Yu et al., 2023) have discovered that it is challenging for the detection models to generalize to

Table 2: Summary of HC-Var

| Task | News | News | News | Review | Review | Writing | QA | QA | QA | QA |
|---|---|---|---|---|---|---|---|---|---|---|
| Topic | World | Sports | Business | IMDb | Yelp | Essay | Finance | History | Medical | Science |
| ChatGPT Vol. | 4,500 | 4,500 | 4,500 | 4,500 | 4,500 | 4,500 | 4,500 | 4,500 | 4,500 | 4,500 |
| Human Vol. | 10,000 | 10,000 | 9,096 | 10,000 | 10,000 | 10,000 | 10,000 | 10,000 | 10,000 | 10,000 |
| Human Src. | XSum | XSum | XSum | IMDb | Yelp | IvyPanda | FiQA | Reddit | MedQuad | Reddit |

unseen language tasks and topics. Different from these existing works, we collect a new dataset to include different varieties to support a comprehensive analysis on their generalization. In Section 5, we discuss potential strategies to overcome the distribution shift. Besides, there are also previous works claiming that the models can struggle to predict texts with shorter lengths (Tian et al., 2023b; Guo et al., 2023). While, our paper finds that it could be related to a poor HC-Alignment (see Section 4.2) and we provide theoretical understandings (Section 4.3) about this issue.

## 3 PRELIMINARY

In this section, we first introduce the details of our proposed dataset, *HC-Var: Human and ChatGPT texts with Variety*. Then we discuss the general experimental setups and evaluation metrics used in the paper. Next, we conduct a preliminary comparison on existing methods under the "in-distribution" setting, before we discuss their generalization behaviors.

### 3.1 HC-VAR: HUMAN AND CHATGPT TEXTS WITH VARIETY

As discussed, we are motivated to study the generalization of ChatGPT detection when faced with various distribution shifts, including *prompts, lengths, topics and language tasks*. Refer to Table 1, existing datasets do not sufficiently support this analysis, because they don't

Table 1: Comparison of Different Datasets

| Dataset | Prompts | Lengths | Topics | Tasks |
|---|---|---|---|---|
| HC-3 (Guo et al., 2023) | ✗ | ✗ | ✓ | ✗ |
| M4 (Wang et al., 2023) | ✗ | ✗ | ✓ | ✓ |
| OGT. (Chen et al., 2023) | ✗ | ✗ | ✗ | ✗ |
| HC-Var (Ours) | ✓ | ✓ | ✓ | ✓ |

cover all types of considered varieties. Therefore, in HC-Var, we create a new dataset, collecting human and ChatGPT generated texts to include these varieties. Overall, as shown in Table 2, the dataset contains 4 different types of language tasks, including news composing (news), review composing (review), essay writing (writing) and question answering (QA). Each task covers 1 to 4 different topics. In HC-Var, human texts are from different public datasets such as XSum, IMDb.[1]

**Variety in Prompts & Lengths.** In each task, we design 3 prompts to obtain ChatGPT outputs to ensure the variety of generated outputs and their lengths. For example, to ask ChatGPT to compose a review for a movie with title <MovieTitle>, we have the prompts:

- *P1:* Write a review for <MovieTitle> in [50, 100, 200] words.
- *P2:* Develop an engaging and creative review for <MovieTitle> in [50, 100, 200] words. Follow the writing style of the movie comments as in popular movie review websites such as imdb.com.
- *P3:* Complete the following: I just watched <MovieTitle>. It is [enjoyable, just OK, mediocre, unpleasant, great]. [It is because that, The reason is that, I just feel that, ...]. [2]

The design of P3 will make ChatGPT texts look much more casual and conversational than P1 and P2 (see Appendix A for some examples). Notably, previous studies (Guo et al., 2023; Kabir et al., 2023) observe that ChatGPT texts are much more formal and official compared with human texts. However, our dataset includes the instances to employ ChatGPT to produce texts, which are casual and close to spoken language. This can greatly enriches the collection of ChatGPT generated outputs. Similarly, under "QA", given a question <Q>, we have the following prompts:

- *P1:* Answer the following question in [50, 100, 150] words. <Q>
- *P2:* Act as you are a user in Reddit or Quora, answer the question in [50,100,150] words. <Q>
- *P3:* Answer the following question in [50, 100, 150] words. <Q> Explain like I am five.

The P3 (which is also used in (Guo et al., 2023)) also encourages the generated answers to be closer to spoken language. Besides, for tasks such as essay writing and news writing where human texts are originally formal, we design various prompts by assigning different writing styles. For example, in essay writing, one of the prompt is "Writing an article with following title like a high school student". More details about the prompt design are in Appendix A.

---

[1] We following existing datasets to take public available datasets as human texts.

[2] Each word / phrase in the gray list has the same chance to be randomly selected. In P3, the each generated text is randomly truncated to 50-200 tokens.

## 3.2 IN-DISTRIBUTION EVALUATION

In this subsection, under our proposed dataset HC-Var, we verify that the training-based detection methods can indeed achieve advantageous detection performance under the "in-distribution" setting, when compared with other detection methods. This part of experiments is also consistent with previous experimental studies (Guo et al., 2023; Chen et al., 2023) which are conducted in other datasets. The extraordinary in-distribution performance motivates us to study its generalization behavior.

**Experimental Setup.** Generally, each experiment is focused on a specified language task, so the detection models are trained and tested on the texts from the same task. For example, under QA, we train the detection models on human and ChatGPT answered questions, and test whether they can distinguish these answers. Under each task, we randomly sample from the datasets to obtain class-balanced training, validation and test subsets (each has an equal number of human and ChatGPT samples). Thus, all training, validation and test datasets contain various topics, prompts and lengths, so distribution shift between training and test set is negligible, namely "in-distribution" evaluation.

**Evaluation Metrics.** We evaluate the detection performance using different metrics: *True Positive Rate (tpr)* shows the detector's power to identify ChatGPT generated texts, *1 - False Positive Rate (1-fpr)* shows the detector's accuracy on human texts, *F1 score* considers the *tpr* and *1-fpr* trade-off. All F1 score, tpr and fpr are calculated under a fixed decision threshold 0.5. We also include *AUROC* which considers all possible thresholds for decision making.

**Performance Comparison.** In Table 3, we report the performance of trained classification models, which are based on model architectures RoBERTa-base, RoBERTa-large and T-5. We also include representative "similarity-based" methods DNA-GPT (Yang et al., 2023) and GPT-PAT (Chen et al., 2023), and "score-based" methods including GLTR (Gehrmann et al., 2019) and GPTZero (GPTZero.com). From the table, we can see that training-based methods outperform non-training based methods under the in-distribution evaluation.

Table 3: In-distribution ChatGPT Detection Performance

|  | News | | | | Review | | | | Writing | | | | QA | | | |
|---|---|---|---|---|---|---|---|---|---|---|---|---|---|---|---|---|
|  | Auc | f1 | tpr | 1-fpr | Auc | f1 | tpr | 1-fpr | Auc | f1 | tpr | 1-fpr | Auc | f1 | tpr | 1-fpr |
| GPTZero | 0.99 | 0.94 | 1.00 | 0.94 | 0.99 | 0.90 | 0.82 | 1.00 | 0.98 | 0.89 | 0.97 | 0.90 | 0.95 | 0.90 | 0.98 | 0.91 |
| GLTR | 0.94 | 0.87 | 0.88 | 0.86 | 0.90 | 0.82 | 0.85 | 0.80 | 0.99 | 0.95 | 0.94 | 0.98 | 0.88 | 0.81 | 0.78 | 0.82 |
| DNA-GPT | 0.92 | 0.90 | 0.89 | 0.89 | 0.93 | 0.90 | 0.88 | 0.89 | 0.97 | 0.92 | 0.88 | 0.95 | 0.87 | 0.82 | 0.86 | 0.80 |
| GPT-PAT | 1.00 | 0.99 | 1.00 | 0.99 | 1.00 | 1.00 | 1.00 | 1.00 | 1.00 | 0.99 | 0.99 | 0.99 | 0.99 | 0.95 | 0.97 | 0.94 |
| RoBERTa-b | 1.00 | 1.00 | 1.00 | 1.00 | 1.00 | 0.99 | 0.99 | 1.00 | 1.00 | 1.00 | 1.00 | 1.00 | 1.00 | 0.98 | 0.99 | 0.98 |
| RoBERTa-l | 1.00 | 1.00 | 1.00 | 1.00 | 1.00 | 0.99 | 1.00 | 0.99 | 1.00 | 1.00 | 1.00 | 1.00 | 1.00 | 0.99 | 0.99 | 0.99 |
| T-5 | 1.00 | 1.00 | 0.99 | 0.99 | 1.00 | 0.99 | 0.99 | 0.99 | 1.00 | 1.00 | 0.99 | 1.00 | 1.00 | 0.98 | 0.98 | 0.96 |

The training-based methods present extraordinary "in-distribution" detection performance. This motivates us to have a further exploration on their generalization performance under out-of-distribution scenarios. In the following, we design experiments to analyze them when the training data cannot fully cover the distribution of test data. Our analysis contains two major scenarios. In Section 4, we consider the scenario that the model trainer aims to detect the texts from their interested language tasks and topics. In this case, the possible distribution shifts can be due to the variation of prompts and lengths. In Section 5, we discuss the cases that the models encounter unforeseen tasks or topics.

# 4 HOW PROMPT & LENGTH AFFECT DETECTION GENERALIZATION

## 4.1 GENERALIZATION TO UNSEEN PROMPTS

To detect ChatGPT texts from a certain language task with several interested topics, it is a realistic and practical scenario that the model trainer collects ChatGPT texts using certain prompts. However, they never know whether there are other unforeseen prompts used to obtain ChatGPT outputs during test. Thus, we aim to analyze how the detection models can generalize to unseen prompts. In detail, refer to Figure 1, we conduct experiment to train the model for multiple trials (in each individual task with the topics in HC-Var). For each task at each time, we train the model on ChatGPT generated texts from one prompt, and test the model on each of three prompts (which we designed in Secion 3) individually. Besides, for each time of training, the human texts are randomly sampled to match the number of generated texts. In Figure 1, we report the F1 score[3] of the trained classifiers. Notably, for these trained models, they have similar (close to 100%) accuracy on human texts (see Appendix B.1). Therefore, these F1-scores are majorly determined by their True Positive Rate, which measure their ability to correctly recognize ChatGPT texts.

---

[3] We report F1 score instead of AUROC, as AUROC considers all thresholds for decision making, which is impractical under unseen distribution shift. All experiments are conducted by 5 times, the average is reported.

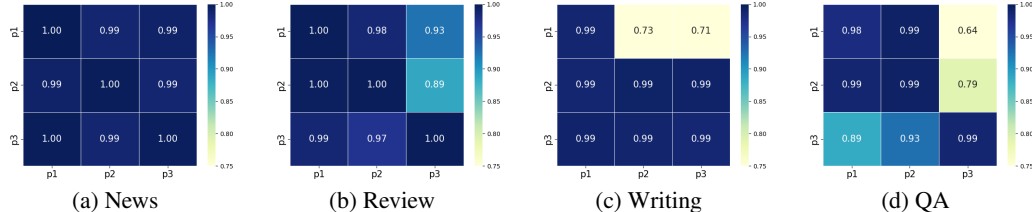

(a) News          (b) Review          (c) Writing          (d) QA

Figure 1: Generalization of RoBERTa-base models among various prompts. Note that each row denotes the prompt during training and each column is the test prompt. F-1 score is reported by color score.

In this section, we study the ChatGPT detection generalization in terms of prompts and lengths under the same topic and domain. Note that we only report the result of a representative model, RoBERTa-base, and results for other models such as RoBERTa-large and T5 are in Appendix B.

**Observations.** From Figure 1, we can observe a great disparity among models that trained and tested on different prompts. For example, under QA, the model trained on P1 or P2 has low F1 scores 0.64 and 0.79 on P3 respectively. While, the model trained on P3 has a better generalization, with F1 score 0.89 and 0.93 on P1 and P2 respectively. Thus, a natural question raises: ***Why does such disparity happen?*** Next, we unveil two potential reasons.

**Reason 1. Prompt Similarity:** Intuitively, the generalization performance can be highly dependent on the "similarity" between the generated texts from two different prompts. In other words, if ChatGPT responds to two prompts in a similar way, it is very likely that the models trained on one prompt can also correctly recognize the texts from the other. Therefore, for two given prompts $P_i$ and $P_j$ (in the same task), we propose the concept of "**prompt similarity**", denoted as $\mathcal{S}(\mathcal{D}_C^{P_i}, \mathcal{D}_C^{P_j})$, which refers to the similarity between the generated texts $\mathcal{D}_C^{P_i}$ and $\mathcal{D}_C^{P_j}$ from prompts $P_i$ and $P_j$. In this work, we calculate this similarity using MAUVE (Pillutla et al., 2021), which is a well-known similarity metric for text distribution, and we report every $\mathcal{S}(\mathcal{D}_C^{P_i}, \mathcal{D}_C^{P_j})$ in Figure 2. In Figure 3, we also visualize the texts from various prompts in the pen-ultimate layer of a pre-trained RoBERTa model. From figures, we can see that the "prompt similarity" has a great impact on generalization. Take QA as an example, the generated texts from P1 and P2 has a high MAUVE similarity 0.97, the representations of texts from P1 and P2 are also correspondingly close to each other. Meanwhile, in Figure 1d, the generalization between P1 and P2 is also high, showing that trained models on similar prompts can well generalize to each other.

**Reason 2. Human-ChatGPT Alignment:** A more interesting study is about generalization between dissimilar prompts. In each task, there are cases where the training and test prompts are not similar but have a good generalization. For example, in review, P1 and P3 are not similar but the model trained on P3 has a high F1 score 0.99 on P1. It suggests that there are other reasons beyond prompt similarity that also affect the generalization performance. In this work, we find: for the training datasets which contain ChatGPT outputs closer to human written texts, the trained model has better generalization. We called this property as the "**Human-ChatGPT (HC) alignment**", which refers to the similarity between $\mathcal{D}_C^{P_i}$ and $\mathcal{D}_H$, and denoted as $\mathcal{S}(\mathcal{D}_C^{P_i}, \mathcal{D}_H)$. In Figure 4a, for each task, we measure HC-alignment for each prompt $P_i$, also using the MAUVE similarity. In Figure 4 (b)-(e), we re-organize the result in Figure 1 using bar plots to show the F1 score of the model trained and tested on each prompt. From the result, we note that the prompts with high "HC Alignment" have better generalization to other prompts. For prompts with low HC-Alignment, they have poorer generalization to other prompts unless they are tested on the prompts with high "prompt similarity" (which we give them a gray color in Figure 4 (b)-(e)). Interestingly, the calculated HC-alignment also reflects our idea during prompt designing in data collection phase. Refer to Section 3.1, in "review" and "QA", P3 is designed to guide the ChatGPT generate texts more conversational in QA and review. From Figure 4a, the HC alignment of P3 is also the highest.

**Insights.** In practice, it is a realistic and reasonable setting to consider multiple and diverse prompts (which we don't include in this discussion, since we only calculate HC-Alignment for each individual prompt). However, our studies draw key insights to bring cautions to the data collection during model training, which is the pitfall of only collecting samples far away from human data. To explain the impact of HC-Alignment on generalization, in Section 4.3, we construct a theoretical analysis to provide deeper understanding. In our discussion, we majorly claim that there can be two types

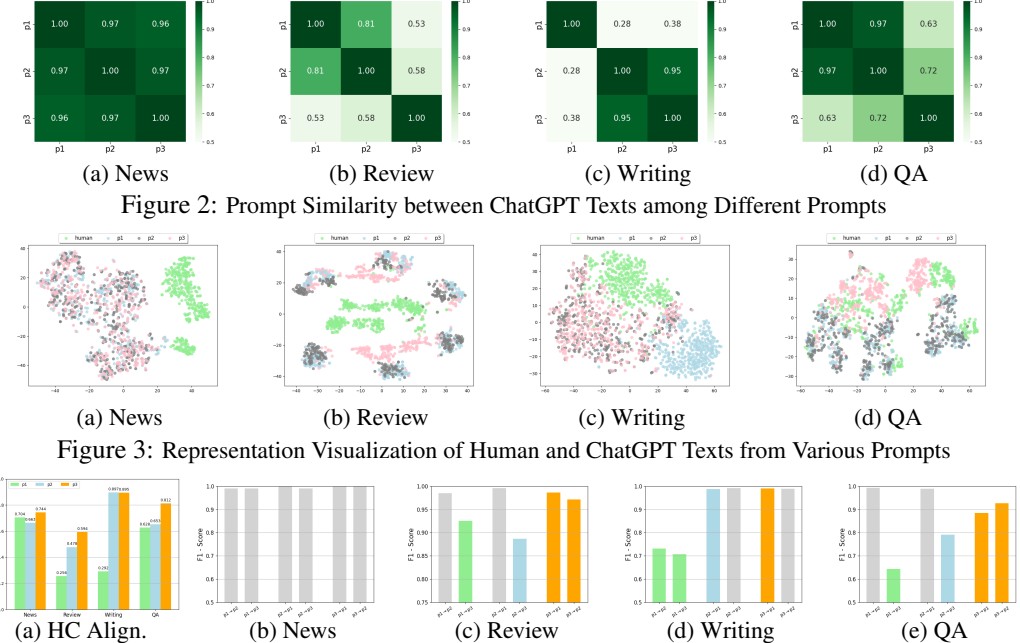

(a) News     (b) Review     (c) Writing     (d) QA

Figure 2: Prompt Similarity between ChatGPT Texts among Different Prompts

(a) News     (b) Review     (c) Writing     (d) QA

Figure 3: Representation Visualization of Human and ChatGPT Texts from Various Prompts

(a) HC Align.     (b) News     (c) Review     (d) Writing     (e) QA

Figure 4: HC-Alignment for different prompts and Generalization.

of factors contributing the HC Alignment. Specifically, for the ChatGPT data $\mathcal{D}_C^{P_i}$ and human data $\mathcal{D}_H$, they can differ in "***ChatGPT Direction***" and "***Irrelevant Direction***" (see Section 4.3 for more details). A larger difference in irrelevant direction can cause the ChatGPT generated texts have a lower HC-Alignment with human texts. Meanwhile, the detection models trained on datasets with low HC-alignment are likely to overfit to this irrelevant direction and suffer from poor generalization. In the next subsection, we provide an example study, to show that the **length of the texts** can be one possible "irrelevant direction" which affects generalization of the models.

## 4.2 GENERALIZATION TO LENGTH SHIFT

Recall that in Section 3, when we design prompts to inquire ChatGPT outputs, we explicitly control the lengths of the generated texts. In this subsection, we show the impact of lengths on the model's generalization. To have an overview on the length distribution of human and ChatGPT texts, in Figure 5a, we plot the density of human texts and ChatGPT texts in HC-Var in one language task "review". Additionally, we include ChatGPT# to show the length distribution if we do not designate the lengths in the inquiries (i.e., by removing "in [50, 100, 200] words" in the prompts). From the Figure 5a, we can see the generated texts from ChatGPT# are much longer compared to human texts. Notably, previous studies (Guo et al., 2023) also find ChatGPT texts are longer than human in their collected QA dataset, HC-3. This suggests the length can be a commonly overlooked factor in previous studies during data collection. (See Appendix B.3 for length comparison in other tasks.)

In our study, we find this difference in length will make a noticeable impact on the trained model's performance. For example, in Figure 5b, we report the performance (TPR, 1-FPR) of the model trained on our dataset when it is tested on samples with various lengths. In Figure 5c, we conduct the same experiment, by replacing the ChatGPT texts in training set to ChatGPT# (without length designation). From the result, we can see the second model struggles on classifying short ChatGPT texts. In other words, the second model tends to predict short ChatGPT texts as human written. A likely reason is that this model is trained to heavily rely on the lengths of the texts for prediction. If a candidate text is short, the model will predict it as human-written. However, text lengths should be an "irrelevant feature" for detection, as ChatGPT can generate shorter or longer texts. In Figure 5b, this issue can be greatly alleviated under our dataset. It may be bcause our collected dataset HC-Var has a much slighter length difference between human and ChatGPT texts (see Figure 5a). This finding encourages us to collect ChatGPT texts to have similar lengths with human texts for training the detection models. It also demonstrates the pitfall if only collecting ChatGPT outputs that are very distinct from human texts. This conclusion echoes back to the discussions in Section 4.1.

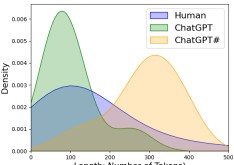 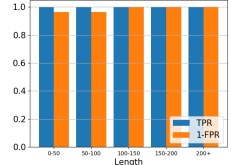 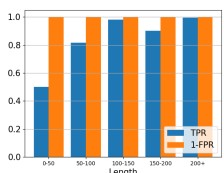

(a) Length Distribution.     (b) Under HC-Var.     (c) No Length Designation.

Figure 5: Impact of Lengths on ChatGPT Detection

## 4.3 THEORETICAL ANALYSIS

In this section, we construct a theoretical model to understand our previous empirical results. We aim to show when the ChatGPT texts and human texts are not well-aligned, it is likely that the model has a poor generalization. To build the connection, our major argument is that the models tend to focus on "***irrelevant directions***" for detection when this alignment is low. In our study, we use a two dimensional toy model with data samples from Gaussian distributions to illustrate our idea.

**Theoretical setup.** We consider a simplified scenario that human texts and ChatGPT texts are lying in a two dimensional data space. As illustrated in Figure 6, we define the $x_1$-axis refers to "*ChatGPT Direction*", which includes principal features to decide whether a sample belongs to human or Chat-GPT. For simplicity, we define the region to the right of line $x_1 = C(C > 0)$ as ChatGPT generated, and we define the left of $x_1 = H(H > 0)$ as human written. Orthogonal to the Chat-GPT direction, we define the $x_2$-axis as "*Irrelevant Direction*". This direction contains features that are irrelevant for ChatGPT detection. Previous discussion in Section 4.2 demonstrates that the length of the texts can be one source of irrelevant features.

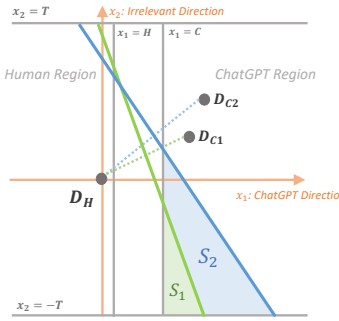

Figure 6: Theorem illustration.

Under this data space, we define the human training data are sampled following a Gaussian distribution $\mathcal{D}_H = \mathcal{N}(0, \sigma^2 I)$. For ChatGPT data, we also assume that they are sampled following a Gaussian distribution in the space $x_1 \geq C$. Recall the previous empirical studies, we find that using different prompts can generate texts with different HC-Alignment. In our analysis, we aim to compare two data collection strategies, with different distances to human data (a.k.a HC Alignment in empirical studies). In detail, we compare the strategies to make samplings from $\mathcal{D}_{C1}$ and $\mathcal{D}_{C2}$:

$$\begin{cases} \mathcal{D}_{C1} = \mathcal{N}(\theta_1, \sigma^2 I), & ||\theta_1||_2 = d, \\ \mathcal{D}_{C2} = \mathcal{N}(\theta_2, \sigma^2 I), & ||\theta_2||_2 = K \cdot d, \end{cases} \quad d \geq C, K > 1 \quad (2)$$

The key difference between the two data distributions is the existence of the term $K$, which decides their distance to the human data. For the centers $\theta_1$ and $\theta_2$, they are uniformly distributed in the ChatGPT region, as long as they have distances $d$ and $K \cdot d$ to the origin. Next, we will study the generalization performance for binary classification models trained on human and ChatGPT texts. Before that, we first define a necessary evaluation metric of model generalization.

**Definition 1** (*False Negative Area*). *For a given model $f$, it could make errors in ChatGPT region under area surrounded by $f$, $x_1 = C$ and $x_2 = \pm T$, where $T > 0$ is a threshold value controlling the limitation of $x_2$. We define the **False Negative Area (FNA)** as the area of the enclosed region.*

As an illustration in Figure 6, $S_1$ and $S_2$ represent the corresponding FNA of $f_1$ and $f_2$, respectively. In our analysis, we denote the FNA of a model $f$ as $\Gamma(f)$. We use it to measure the models' error rate on unforeseen ChatGPT generated data, which are not covered by the collected training data. Next, we formally state our main theory by analyzing the FNA of the models $f_1$ and $f_2$:

**Theorem 1.** *Given the human training data $\mathcal{D}_H$, ChatGPT training data $\mathcal{D}_{C1}$, $\mathcal{D}_{C2}$. For two classifiers $f_1$ and $f_2$ which are trained to minimize the error under a class-balanced dataset:*

$$f_i = \arg\min_f Pr.(f(x) \neq y), \quad where \begin{cases} x \sim \mathcal{D}_{Ci}, & if \ y = 1 \\ x \sim \mathcal{D}_H, & if \ y = 0 \end{cases}$$

*Suppose the maximal FNA that $f_1$ can achieve is denoted as $\sup \Gamma(f_1)$. Then, with probability at least $\left(1 - \left(\frac{\pi}{2} - \frac{C}{d} + \Omega(\frac{C}{d})^3\right)/\left(\frac{\pi}{2} - \frac{C}{Kd}\right)\right)$, we have the relation:*

$$\left(\frac{\Gamma(f_2)}{\sup \Gamma(f_1)}\right)^2 \geq \left(1 + (K-1) \cdot \frac{1}{1 + 2T \cdot \Omega(1/d)}\right) > 1. \tag{3}$$

The proof is deferred to Appendix C. This theorem suggests that the FNA of $f_2$ is likely to be larger than the worst case of $f_2$ (with a moderate probability), since their FNA ratio is larger than 1. Moreover, both the probability term and the FNA ratio term (Eq.5) are monotonically increasing with the term $K$. It suggests the larger $K$ it is, the higher chance of $f_2$ can have a poorer generalization than $f_1$. Refer to Figure 6, compared with $f_1$, the model $f_2$ has a larger FNA, because its decision boundary has a smaller slope, which means $f_2$'s prediction is more relied on the irrelevant direction.

## 5 GENERALIZATION OF CHATGPT DETECTION ACROSS TOPICS & DOMAINS

In this section, we discuss the circumstances that the models can face texts from unforeseen language tasks or topics. Under this setting, we find the trained models can also extract useful features to help the generalization to other unforeseen tasks or topics, which we called "**transferable features**". We also validate one frequently applied strategy, transfer learning (Pan & Yang, 2009), can be benefited from this property. Notably. in this section, we only provide the results for task-level generalization, and we leave the topic-level study in Appendix B, where we can draw similar conclusions.

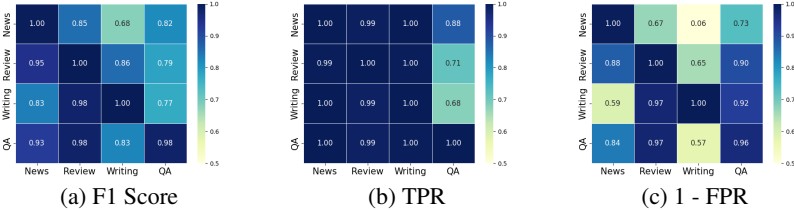

(a) F1 Score      (b) TPR      (c) 1 - FPR

Figure 7: Generalization of RoBERTa-base model across Various Language Tasks

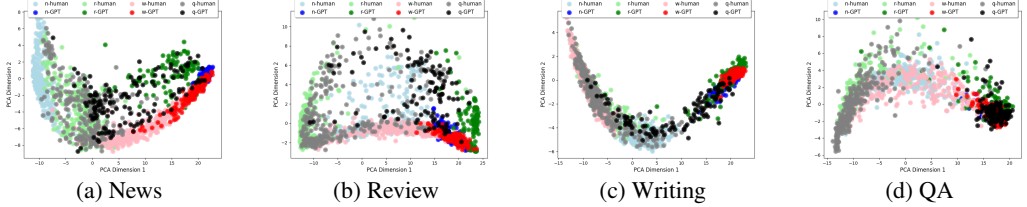

(a) News      (b) Review      (c) Writing      (d) QA

Figure 8: Representation space visualization on models trained on each task

### 5.1 GENERALIZATION ACROSS TOPICS & DOMAINS

In this subsection, we conduct experiments to test the RoBERTa-base classification method's generalization across language tasks (and topics). In particular, in Figure 7, we train the model on the human and ChatGPT texts from each language task individually and we check whether it can correctly classify texts from other tasks. Since these tasks have different number of samples in HC-Var, we randomly sample 4,000 ChatGPT and 4,000 human samples for training in all experiments. In each training set, the ChatGPT texts will contain various topics (if exist) and various prompts.

In the experiments, we report the evaluation metrics including F1-score, TPR and 1-FPR. Based our reported results in Figure 8, we can see that the trained models will have a performance drop on either human texts or ChatGPT texts. For example, the model trained on "writing" cannot effectively detect the ChatGPT generated texts in "QA". Similarly, the models trained on "news" can hardly recognize human written texts in "writing". This result shows that models could make errors on both human or ChatGPT texts. In Appendix B, we provide the results for topic-level generalization, where we can draw similar conclusions. In reality, due to the versatility of ChatGPT to handle various tasks, it is infeasible to collect texts from all possible tasks for model training.

### 5.2 FINE-TUNING WITH A FEW SAMPLES HELPS CROSS-DOMAIN / TOPIC DETECTION

In this part, we identify a ponteitial way to improve the ChatGPT detection in the unforeseen tasks (or topics). It is based on our finding that the models trained in each individual task can learn

helpful features for other tasks. As an evidence, in Figure 8, we visualize the learned representations for various tasks rendered by the trained model in Section 5.1. From these figures, we note that the ChatGPT and human texts from unseen tasks during training are also well-separated in the representation space. It demonstrates the models can indeed learn useful features which are helpful to distinguish human and ChatGPT texts in other domains, so we call them "***Transferable Features***".

Table 4: Transfer Learning (Task-level) Performance via Linear Probing and Fine-Tuning

| Target | news | | | review | | | writing | | | QA | | |
|---|---|---|---|---|---|---|---|---|---|---|---|---|
| | r→n | w→n | q→n | n→r | w→r | q→r | n→w | r→w | q→w | n→q | r→q | w→q |
| No Transfer | 0.946 | 0.835 | 0.927 | 0.854 | 0.980 | 0.981 | 0.681 | 0.858 | 0.827 | 0.819 | 0.789 | 0.771 |
| LP-5 | 0.991 | 0.990 | 0.972 | 0.901 | 0.958 | 0.987 | 0.901 | 0.967 | 0.902 | 0.772 | 0.860 | 0.849 |
| FT-5 | 0.952 | 0.923 | 0.932 | 0.965 | 0.952 | 0.940 | 0.871 | 0.898 | 0.835 | 0.848 | 0.893 | 0.869 |
| LP-Scratch-5 | $0.959 \pm 0.019$ | | | $0.839 \pm 0.057$ | | | $0.871 \pm 0.024$ | | | $0.697 \pm 0.082$ | | |
| FT-Scratch-5 | $0.946 \pm 0.033$ | | | $0.925 \pm 0.033$ | | | $0.867 \pm 0.021$ | | | $0.687 \pm 0.047$ | | |
| LP-10 | 0.993 | 0.992 | 0.993 | 0.938 | 0.986 | 0.984 | 0.916 | 0.971 | 0.934 | 0.839 | 0.887 | 0.859 |
| FT-10 | 0.978 | 0.978 | 0.983 | 0.951 | 0.968 | 0.967 | 0.936 | 0.956 | 0.936 | 0.870 | 0.913 | 0.909 |
| LP-Scratch-10 | $0.979 \pm 0.005$ | | | $0.934 \pm 0.013$ | | | $0.906 \pm 0.023$ | | | $0.764 \pm 0.071$ | | |
| FT-Scratch-10 | $0.983 \pm 0.006$ | | | $0.941 \pm 0.020$ | | | $0.939 \pm 0.018$ | | | $0.778 \pm 0.051$ | | |

We use the blue color to highlight the case that transfer learning outperforms training from scratch.

To further verify the existence of transferable features, we conduct experiments to investigate transfer learning (Hendrycks et al., 2019) for domain adaption. In reality, if the model trainer encounters test samples from the language tasks (or topics) which are not involved in the training set, it is a practical and feasible solution for them to collect several samples in the same task as the test sample by themselves. Therefore, in our study, we consider two types of transfer learning strategies: *Linear Probing (LP)*, which refers to the strategy that only the linear classifier (based on extracted features) is optimized; and *Fine Tuning (FT)* which refers to the strategy that all layers are optimized.

In our experiment, we consider there are 5 and 10 more samples from both human data and ChatGPT texts are sampled for fine-tuning the models. In Table 4, we report the tuned models performance (F1 score) when tested on different targeted (downstream) tasks from various source models. For example, "$r \rightarrow n$" means the model transferred from "review" for a downstream task "news". Besides, we also include the original performance (from Figure 7), which is the performance before transfer learning (denoted as "No Transfer" in Table 4). For comparison, we report the result if these models are tuned from scratch (on pre-trained RoBERTa-base model without training for detection).

From the result, we can see transfer learning can benefit the detection performance in general. For example, when compared with "No Transfer", linear probing (LP) or Fine-tuning (FT) can improve the downstream task performance in most cases (except for $w \rightarrow r$ with 5 training samples). Moreover, when compared to the models training from scratch, the transferred models also achieve higher performance in all considered language tasks. It suggests that those pre-trained models can offer helpful features beyond the collected data samples for down-stream tuning. These results indeed show that there are shared features, a.k.a, transferable features, which are generally useful to distinguish human and ChatGPT texts in various domains. Remarkably, in Section 4.3, we introduce the notion "ChatGPT Direction", which contains the fundamental and principal features to distinguish human and ChatGPT texts. Ideally, these features should be universally helpful for ChatGPT detection in all tasks and topics. However, it is hard to verify their existence in reality, because of the difficulty to consider all possible topics and tasks that ChatGPT can handle. Thus, we use "transferable features" to refer the shared features only in our studied topics and tasks.

## 6    CONCLUSION AND LIMITATION

**Conclusion**: In this paper, we conduct a comprehensive analysis on the generalization behavior of training-based ChatGPT detection methods. Due to the limitation of existing datasets, we collect a new dataset HC-Var, with various types of ChatGPT generated texts and human texts. Our empirical and theoretical studies draw key findings on factors which affect the generalization. We provide insights on the data collection and domain adaption strategy for ChatGPT detection.

**Limitation**: There also other factors which can influence the detection that are not discussed. For example, we have not investigate the scenario that the texts are composed by other language models, such as LLaMA2 (Touvron et al., 2023), or first generated by ChatGPT and then manipulated (i.e., rephrased) by other language models. It is also likely that a given candidate text is partially written by ChatGPT. Besides, as a foundation model, ChatGPT can achieve much more language tasks, such as programming code writing (Tian et al., 2023a). In this work, our major scope and objective is to provide a comprehensive analysis on the generalization of those training-based detectors.

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

# A  DESIGN OF DATASET

## A.1  PROMPTS

In this part, we provide the details of our prompt design.

For news, with a news summary <Summary> from AG-News Dataset, we consider the prompts:

- *P1:* Write a [150, 300] words article following the summary: <Summary>
- *P1:* Write a [150, 300] words article like a commentator following the summary: <Summary>
- *P1:* Write a [150, 300] words article like a journalist following the summary: <Summary>

For Review, with a movie title <MovieTitle>, we consider the prompts:

- *P1:* Write a review for <MovieTitle> in [50, 100, 200] words.
- *P2:* Develop an engaging and creative review for <MovieTitle> in [50, 100, 200] words. Follow the writing style of the movie comments as in popular movie review websites such as imdb.com.
- *P3:* Complete the following: I just watched <MovieTitle>. It is [enjoyable, just OK, mediocre, unpleasant, great]. [It is because that, The reason is that, I just feel that, ...]. [4]

For writing, with a essay topic <EssayTitle>, we consider the prompts:

- *P1:*Write a [200, 300] words essay like a novelist with the following title: <EssayTitle>.
- *P2:*Write a [200, 300] words essay with the following title: <EssayTitle>.
- *P3:*Write a [200, 300] words essay like a high school student with the following title: <EssayTitle>.

For QA, with a question <Q>, we consider the prompts:

- *P1:* Answer the following question in [50, 100, 150] words. <Q>
- *P2:* Act as you are a user in Reddit or Quora, answer the question in [50,100,150] words. <Q>
- *P3:* Answer the following question in [50, 100, 150] words. <Q> Explain like I am five.

---

[4]Each word / phrase in the gray list has the same chance to be randomly selected. In P3, the each generated text is randomly truncated to 50-200 tokens.

A.2 EXAMPLES

In this part, we also provide example texts from both human and ChatGPT, in review (for the movie "Interstellar"), and QA (for the question "Buying a house, how much should my down payment be? "). We include the examples about news and writing because they are longer. From these example, we find that the human texts can be indeed more conversaional and informal than ChatGPT texts. Some of our prompts like "P3" in reviews and ChatGPT can successfully make the generated texts to be more conversational and casual.

Table 5: Example for Human Texts and ChatGPT texts in IMDb Review for movie "Interstellar"

| Class | Prompt | Texts |
|---|---|---|
| Human | N/A | I think just about everything has been said about this film now. But, I can still tell you what this masterpiece is to me. To me, this movie is possibly the most relevant movie ever, because it questions our own humanity relative to the Universe. Whether that's our ability to love, think, or persevere and walk into the unknown. We are explorers, and curious at heart. This untameable curiosity is not our end, but our beginning. It is what advanced this civilization and it will continue to do so. So never, never let anybody tell you that we shouldn't look towards the stars and wonder, because that's what makes us human. Without this stargazing we are merely animals, accepting our fate in the dust. |
| ChatGPT | P1 | "Interstellar" is a cinematic masterpiece that transcends time and space. Christopher Nolan's visionary direction, paired with Hans Zimmer's haunting score, delivers an epic journey through the cosmos. The film's scientific accuracy and emotional depth are nothing short of awe-inspiring. Matthew McConaughey's performance as Cooper is a tour de force, grounding the story's grandeur with genuine humanity. The visual effects are breathtaking, immersing viewers in wormholes and distant planets. While the plot can be complex, it rewards those willing to delve into its intricate layers. "Interstellar" is a thought-provoking, visually stunning odyssey that will leave you pondering the mysteries of the universe long after the credits roll. |
| ChatGPT | P2 | "Interstellar" is a mind-bending journey through the cosmos that will leave you awestruck and pondering the mysteries of the universe long after the credits roll. Christopher Nolan's visionary direction takes us on a rollercoaster ride through wormholes, black holes, and the very fabric of time and space. Matthew McConaughey delivers a stellar performance as Cooper, a father willing to sacrifice everything for the survival of humanity. Hans Zimmer's haunting score adds an emotional depth that tugs at your heartstrings. With stunning visuals and a thought-provoking narrative, "Interstellar" is a cinematic masterpiece that transcends space and time to become an unforgettable exploration of human resilience and the power of love. |

| Class | Prompt | Texts |
|---|---|---|
| ChatGPT | P3 | The movie's exploration of space and time has left me with a sense of awe and wonder. Christopher Nolan's direction and Hans Zimmer's hauntingly beautiful score created a cinematic experience that was truly immersive. The way the film delves into complex scientific concepts like black holes, time dilation, and the theory of relativity was both mind-boggling and intellectually stimulating. It's one of those rare movies that not only entertains but also challenges your understanding of the universe. The emotional depth of the story, particularly the bond between Cooper (Matthew McConaughey) and his daughter Murph (Jessica Chastain and Mackenzie Foy), tugged at my heartstrings. Their relationship, which spans across time and space, added a profound layer to the film's narrative. |

Table 6: Example for Human Texts and ChatGPT texts in QA. The question is "Buying a house, how much should my down payment be?"

| Class | Prompt | Texts |
|---|---|---|
| Human | N/A | How much should my down payment be? Ideally 20% of the purchase price because with 20% of the purchase price, you don't have to pay a costly private mortgage insurance (PMI). If you don't have 20% down and come across a good property to purchase, it is still a good idea to go forward with purchasing with what you are comfortable with, because renting long term is generally never a good idea if you want to build wealth and become financially independent. How much should I keep in my emergency fund? People say 3-12 months of living expenses. Keep in mind though, in most cases, if you lose your job, you are entitled to unemployment benefits from the government. How long should my mortgage be? 30 year amortization is the best. You can always opt to pay more each month. But having that leverage with a 30 year loan can allow you to invest your savings in other opportunities, which can yield more than mortgage interest. Best of luck!. |
| ChatGPT | P1 | The ideal down payment for buying a house typically ranges from 10% to 20% of the home's purchase price. A 20% down payment is often recommended because it helps avoid private mortgage insurance (PMI), reduces monthly mortgage payments, and demonstrates financial stability to lenders. However, some loan programs allow for lower down payments, such as 3.5% for FHA loans or 5% for conventional loans. The specific amount depends on your financial situation, credit score, and the housing market. A higher down payment can offer financial security and lower long-term costs, but it's essential to balance it with your overall financial goals and budget. |
| ChatGPT | P2 | For a down payment on a house, aim for 20% of the home's purchase price. This percentage helps you avoid private mortgage insurance (PMI), which can save you money in the long run. However, some programs allow as low as 3.5% or 5%, like FHA or conventional loans. The ideal amount depends on your financial situation and goals. A higher down payment offers financial security, while a lower one may be more accessible upfront. Balance your budget and future plans to decide what works best for you. |

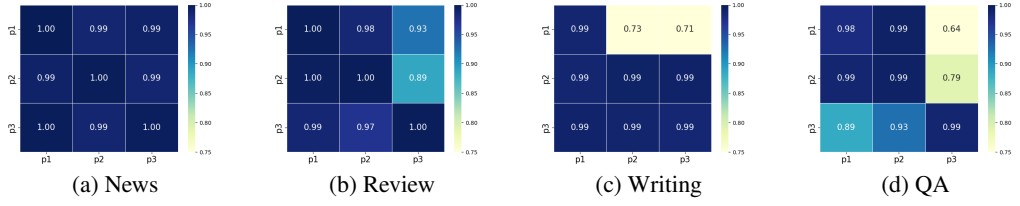

Figure 9: Generalization of **RoBERTa-base models** among various prompts.

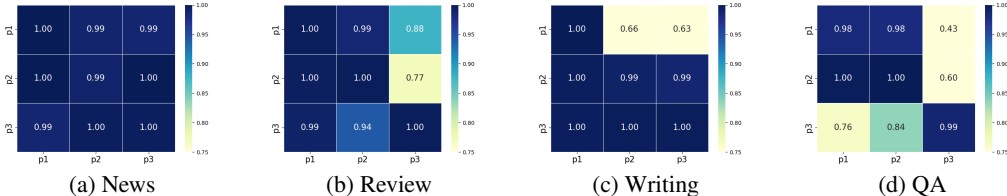

Figure 10: **TPR** Generalization of RoBERTa-base models among various prompts.

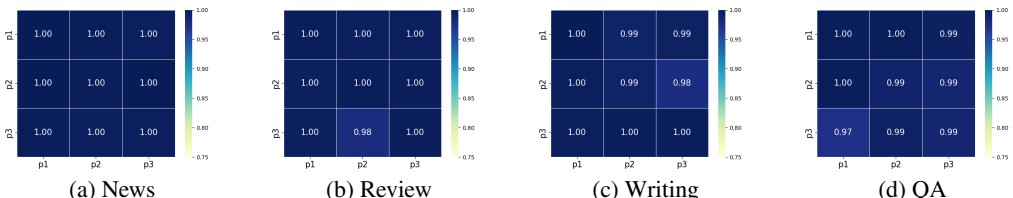

Figure 11: **1 - FPR** Generalization of RoBERTa-base models among various prompts.

| ChatGPT | P3 | When you want to buy a house, you need to save some money to give to the people who are selling the house. This money is called a "down payment." Imagine it's like when you want to trade your toys with a friend; you might need to give them a few of your toys to get their cool toy. Usually, grown-ups suggest giving about 20% of the house's price as a down payment. So, if the house costs $100, you'd give $20 as a down payment. This helps because the more you give upfront, the less money you need to borrow (like borrowing toys), and it can also make it easier to get a loan to buy the house. But remember, it can vary, so talk to a grown-up or a financial expert to decide what's best for you. |
| --- | --- | --- |

# B ADDITIONAL EXPERIMENTS

In this part, we provide additional experimental results which we mentioned in the main text. In Section B.1, we provide discussions on the reason we choose F1-score as the universal metric in Section 4.1. In Section B.2, we analyze the generalization performance for various model architectures. In Section B.3, we include the whole results about the study of generalization on lengh distribution shift. In Section B.4, we provide the transfer learning results about topic-level generalization.

## B.1 ADDITIONAL RESULTS ABOUT SECTION 4.1

In the main text in Section 4.1, we choose F1-score as the standard for model performance evaluation. We also mentioned that the all trained models (under various prompts) have a similar False Positive Rate. Therefore, the True Positive Rate decides the F1-score. In this part, to support our claims, we provide the complete results in Figure 9, Figure 10 and Figure 11. From the results, we can find all the models have similar (1-fpr) in all considered tasks.

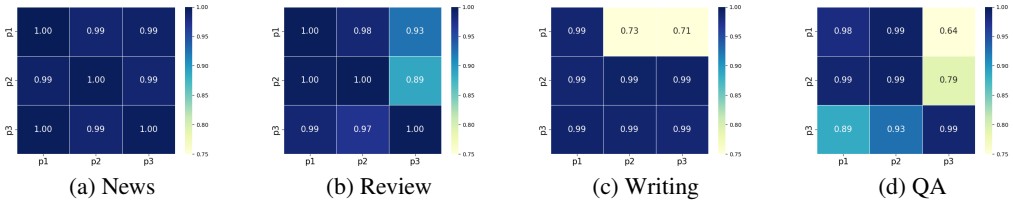

Figure 12: Generalization of **RoBERTa-base models** among various prompts.

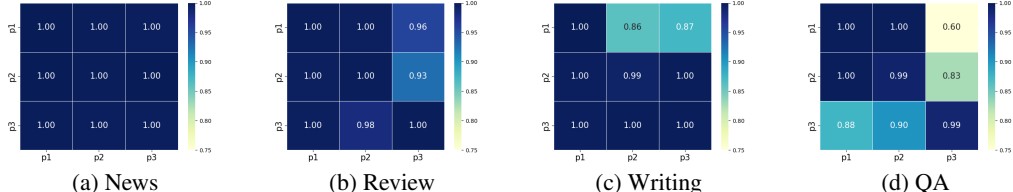

Figure 13: Generalization of **RoBERTa-large models** among various prompts.

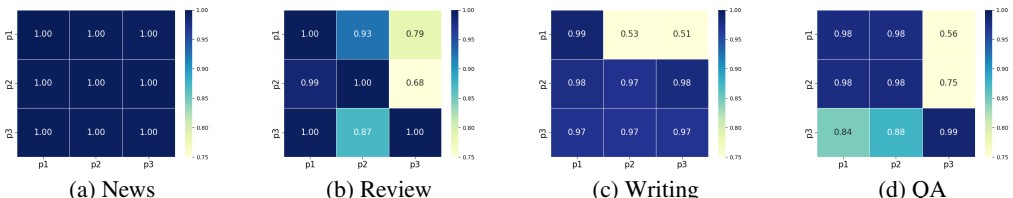

Figure 14: Generalization of **T-5-base models** among various prompts.

**How is the detection when training with multiple prompts?** We conduct an experiment similar to the study in Figure 1, by training with multiple prompts. In detail, for each language tasks, we consider to train the detection model on ChatGPT texts from 2 prompts, and we test the trained model on each of the 3 prompts. The table below reports the F-1 score of the detection models.

| News | P1 | P2 | P3 | Review | P1 | P2 | P3 |
|------|------|------|------|--------|------|------|------|
| P1&P2 | 1.00 | 1.00 | 1.00 | P1&P2 | 1.00 | 1.00 | 0.94 |
| P2&P3 | 1.00 | 1.00 | 1.00 | P2&P3 | 1.00 | 1.00 | 0.99 |
| P1&P3 | 1.00 | 1.00 | 1.00 | P1&P3 | 1.00 | 1.00 | 1.00 |

| Writing | P1 | P2 | P3 | QA | P1 | P2 | P3 |
|---------|------|------|------|----|------|------|------|
| P1&P2 | 1.00 | 1.00 | 0.99 | P1&P2 | 1.00 | 1.00 | 0.84 |
| P2&P3 | 1.00 | 1.00 | 1.00 | P2&P3 | 1.00 | 1.00 | 0.98 |
| P1&P3 | 0.99 | 1.00 | 1.00 | P1&P3 | 1.00 | 0.99 | 0.99 |

From the results, we can have the following observations: (1) for a test prompt appearing in the training dataset, the model has a high detection performance on this prompt. (2) Under the task "Review" and "QA", if we train the model using ChatGPT texts from "P1 & P2", the model has a relatively lower detection generalization to P3. This observation is consistent to our major claim in the analysis in Section 4.1. In Section 4.1, we claim: for model trained on prompts with low HC-Alignment, they may have poorer generalization. Notably, in "Review" and "QA", both P1 and P2 has lower HC-Alignment than P3 (see Figure 4 (a)). Thus, training on them together might also be insufficient to correctly detect texts from P3. Thus, we see a lower detection performance in the table above.

### B.2 ADDITIONAL RESULTS ON OTHER MODELS

In Figure 12, Figure 13 and Figure 14, we conduct experiments to repeat the similar study as in Section 4.1, under different model architectures, RoBERTa-large and T5-base. From the results, we can see that these two models share a similar generalization behavior as RoBERTa-base, which is majorly discussed in the main text. This result suggest that the data distribution make a significant influence on the model generalization. However, the generalization performance between the mod-

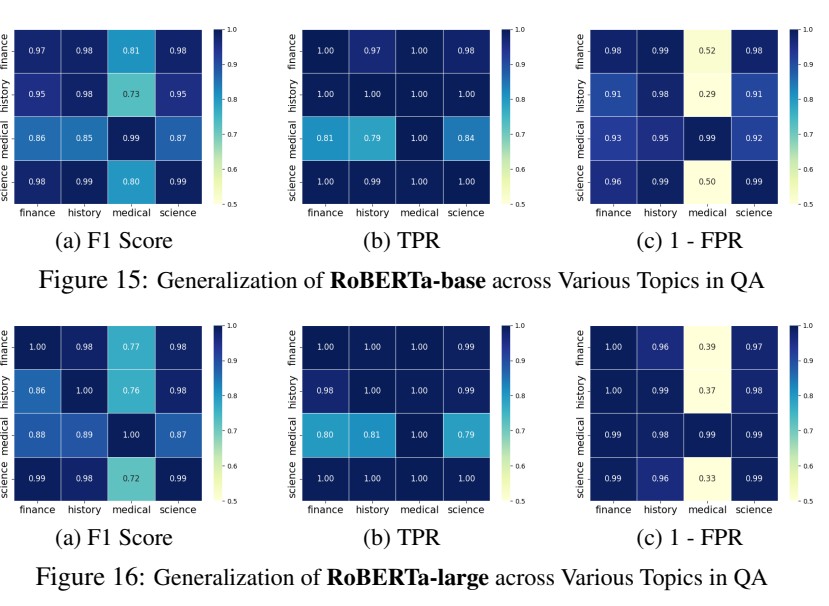

Figure 15: Generalization of **RoBERTa-base** across Various Topics in QA

Figure 16: Generalization of **RoBERTa-large** across Various Topics in QA

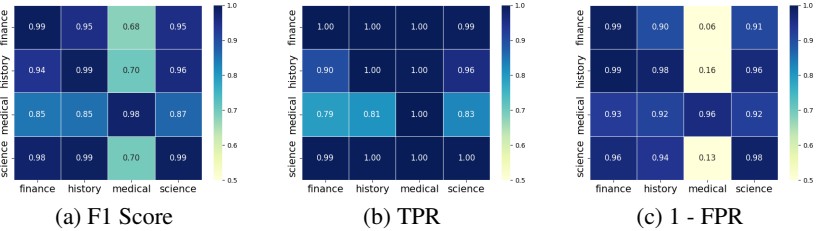

Figure 17: Generalization of **t5-base** across Various Topics in QA

els is also slightly different, for example, RoBERTa-large models tend to have higher performance than T5-base, which suggests that the model architecture can also make a differene. In Figure 15, Figure 16 and Figure 17, we conduct a similar study for topic-level generalization. In Figure 18, Figure 19 and Figure 20, we conduct a study for task-level generalization. These results can consistently validate our analysis in the main context, regardless of the model architecture.

### B.3 DETAILED RESULTS FOR LENGTHS

In Section 4.2, we only report the study about the impact from length in one task, "review". In this part, we provide the complete results for all tasks in HC-Var. In Figure 21, Figure 22 and Figure 23, we provide the same results as in Section 4.2, where we compare the length distribution of human texts and ChatGPT texts (with and without length designation). From the results, we can see that: in Review and QA, our collected dataset has a better alignment with human texts, compared to ChatGPT#. They meanwhile have a better performance especially on shorter texts. For news and writing. As a result, there is a negligible impact from controlling the length, because all human and ChatGPT texts are long texts.

### B.4 ADDITIONAL RESULTS ON TOPIC LEVEL TRANSFERABILITY

In Figure 24 and 15, we conduct experiments to demonstrate the "topic-level" generalization for RoBERTa-base detection model, and its latent space feature visualization. From the results, we can also see that the original models may face performance drop on either human or ChatGPT textst. However, the feature representations of the samples in unforeseen tasks are also well-separated, which is similar to the analysis about "task-level" generalization in Figure 7. Moreover, in Table 7, we conduct an experiment using transfer learning, similar to Table 4. From the table, we can also see that the pre-trained models can significantly help the downstream tasks.

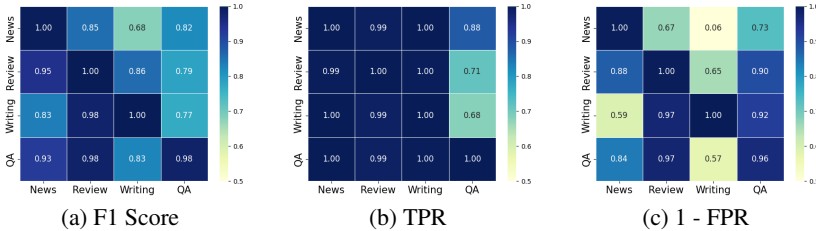

Figure 18: Generalization of **RoBERTa-base** across Various Tasks

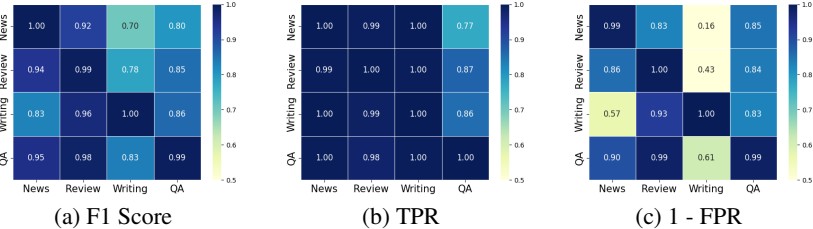

Figure 19: Generalization of **RoBERTa-large** across Various Tasks

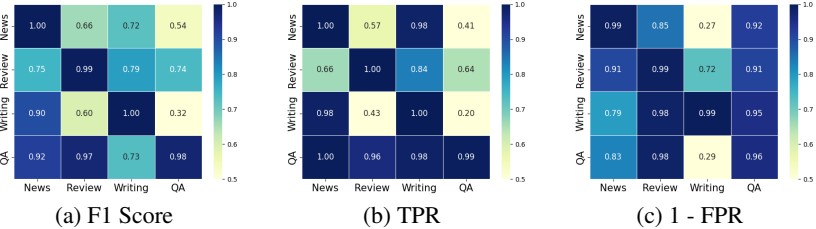

Figure 20: Generalization of **t5-base** across Various Tasks

Table 7: Transfer Learning (Topic-level) Performance via Linear Probing and Fine-tuning in QA

| Target | finance | | | history | | | medical | | | science | | |
|---|---|---|---|---|---|---|---|---|---|---|---|---|
| | h→f | m→f | s→f | f→h | m→h | s→h | f→m | h→m | s→m | f→s | h→s | m→s |
| No Transfer | 0.953 | 0.859 | 0.977 | 0.982 | 0.852 | 0.991 | 0.809 | 0.729 | 0.802 | 0.979 | 0.955 | 0.873 |
| LP-5 | 0.970 | 0.881 | 0.971 | 0.989 | 0.889 | 0.991 | 0.942 | 0.923 | 0.930 | 0.980 | 0.978 | 0.875 |
| FT-5 | 0.952 | 0.904 | 0.945 | 0.940 | 0.952 | 0.958 | 0.873 | 0.836 | 0.840 | 0.931 | 0.925 | 0.879 |
| LP-Scratch-5 | $0.816 \pm 0.025$ | | | $0.813 \pm 0.027$ | | | $0.786 \pm 0.057$ | | | $0.710 \pm 0.068$ | | |
| FT-Scratch-5 | $0.820 \pm 0.034$ | | | $0.806 \pm 0.037$ | | | $0.786 \pm 0.084$ | | | $0.677 \pm 0.097$ | | |
| LP-10 | 0.976 | 0.906 | 0.974 | 0.988 | 0.901 | 0.992 | 0.942 | 0.926 | 0.936 | 0.982 | 0.976 | 0.886 |
| FT-10 | 0.965 | 0.935 | 0.969 | 0.962 | 0.951 | 0.976 | 0.894 | 0.852 | 0.864 | 0.946 | 0.927 | 0.898 |
| LP-Scratch-10 | $0.859 \pm 0.029$ | | | $0.844 \pm 0.025$ | | | $0.792 \pm 0.060$ | | | $0.798 \pm 0.025$ | | |
| FT-Scratch-10 | $0.891 \pm 0.049$ | | | $0.901 \pm 0.035$ | | | $0.865 \pm 0.046$ | | | $0.834 \pm 0.055$ | | |

## C   THEORY PROOFS

In this section, we provide the detailed proofs for our theoretical study. Recall the discussion in Section 1, we aim to compare the strategies to make samplings from $\mathcal{D}_{C1}$ and $\mathcal{D}_{C2}$:

$$\begin{cases} \mathcal{D}_{C1} = \mathcal{N}\left(\theta_1, \sigma^2 I\right), \quad ||\theta_1||_2 = d, \\ \mathcal{D}_{C2} = \mathcal{N}\left(\theta_2, \sigma^2 I\right), \quad ||\theta_2||_2 = K \cdot d, \end{cases} \quad d \geq C, K > 1 \tag{4}$$

**Theorem 1.** *Given the human training data $\mathcal{D}_H$, ChatGPT training data $\mathcal{D}_{C1}$, $\mathcal{D}_{C2}$. For two classifiers $f_1$ and $f_2$ which are trained to minimize the error under a class-balanced dataset:*

$$f_i = \arg\min_f Pr.(f(x) \neq y), \quad where \begin{cases} x \sim \mathcal{D}_{Ci}, & if \ y = 1 \\ x \sim \mathcal{D}_H, & if \ y = 0 \end{cases}$$

*Suppose the maximal FNA that $f_1$ can achieve is denoted as $\sup \Gamma(f_1)$. Then, with probability at least $\left(1 - \left(\frac{\pi}{2} - \frac{C}{d} + \Omega(\frac{C}{d})^3\right)/\left(\frac{\pi}{2} - \frac{C}{Kd}\right)\right)$, we have the relation:*

$$\left(\frac{\Gamma(f_2)}{\sup \Gamma(f_1)}\right)^2 \geq \left(1 + (K-1) \cdot \frac{1}{1 + 2T \cdot \Omega(1/d)}\right) > 1. \tag{5}$$

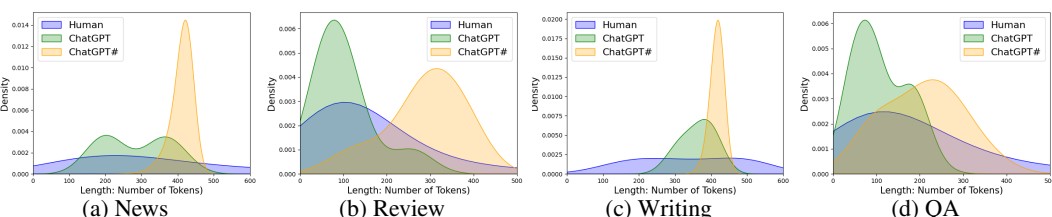

Figure 21: Length Distribution in Human, ChatGPT (in HC-Var), and ChatGPT#

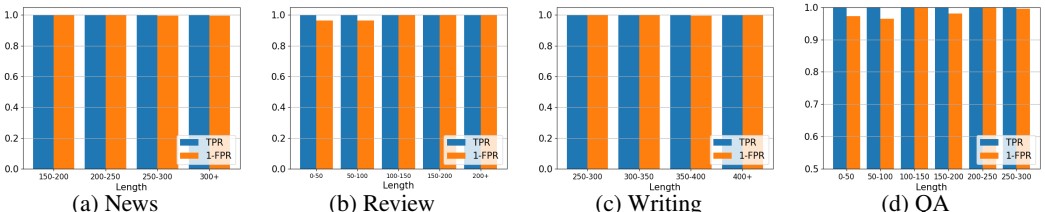

Figure 22: TPR and 1-FPR of RoBERTa-base model **trained under ChatGPT from HC-Var**.

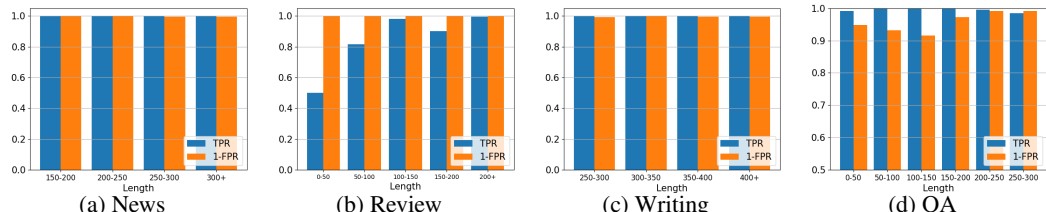

Figure 23: TPR and 1-FPR of RoBERTa-base model **trained under ChatGPT#**.

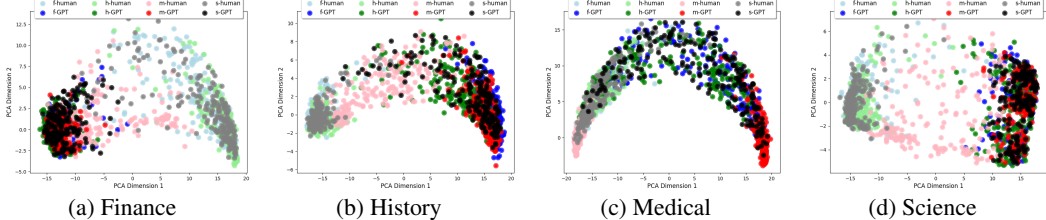

Figure 24: Representation space visualization on models trained on each topic (in QA)

*Proof.* We first aim to find the worst model that $f_1$ can achieve largest FNA. To achieve this goal, we first define its center $\theta_1$ has a location $(h, a)$ where $h^2 + a^2 = d^2$, $h \geq C$. We can suppose $a \geq 0$ without loss of generality since the data space is symmetric on $x_2 = 0$. Therefore, given two classes for classification, with the negative class $\mathcal{D}_H = \mathcal{N}(0, \sigma^2 I)$ and positive class: $\mathcal{D}_{C1} = \mathcal{N}((h, a), \sigma^2 I)$, we can find the optimal classifier $f$ has a decision boundary which is orthogonal to this line that passes $(0, 0)$ and $(h, a)$ and passes their center point $(\frac{h}{2}, \frac{a}{2})$. In specifics, we can get the expression of $f_1$'s decision boundary:

$$l_1 : y = -\frac{h}{a}x + \frac{a^2 + h^2}{2a} \tag{6}$$

Next, we will show that this model $f_1$ will achieve the worst case with largest FNA (the area of the region enclosed by $l_1$, $x_2 = C$ and $x_1 = \pm T$), when $h = C$. To calculate the area of the enclosed region (which is a triangle), we find it has a tall and height:

$$\text{Tall} : -\frac{d^2 - a^2}{a} \cdot C + \frac{d^2}{2a} - C, \quad \text{Height} : \frac{d^2}{2h} + \frac{\sqrt{d^2 - h^2}}{h} \cdot T - T \tag{7}$$

It is easy to see the tall is monotonously increasing as $a$ increases, and height is monotonously decreasing as $h$ increases (by calculating their derivatives). This fact suggests that the tall is also a decreasing function for $h$, as $a$ and $h$ has the relation $a^2 + h^2 = d^2$. Therefore, FNR of $f_1$, which is decided by the multiplication of tall and height, is a decreasing function in terms of $h$. Given that $h \leq C$, the worst case is achieved when $h = C$. Under $h = C$, the model $f_1$ has an FNR:

$$\sup \Gamma(f_1) = \frac{(B^2 + 2BT - C^2)^2}{8BC}, \quad \text{where } B^2 + C^2 = d^2. \tag{8}$$

Next, we discuss one special case about model $f_2$, which is denoted as $f_2^*$. Then, we calculate the probability that other possible $f_2$ is worse than this specific case $f_2^*$. In detail, we consider that $\theta_2 = K \cdot \theta 1$, which means they are in the same direction from origin. For the model $f_2^*$, we can calculate the model for this special case of $f_2^*$:

$$y = -\frac{C}{B} + K \cdot \frac{B^2 + C^2}{2B} \tag{9}$$

and it has an FNR:

$$\Gamma(f_2^*) = \frac{(2BT + KB^2 + (K-2)C^2)^2}{8BC} \tag{10}$$

Next, we calculate the ratio between $\Gamma(f_2^*)$ and $\sup \Gamma(f_1)$:

$$\left(\frac{\Gamma(f_2^*)}{\sup \Gamma(f_1)}\right)^2 = \frac{2BT + KB^2 + (K-2)C^2}{2BT + B^2 - C^2} = 1 + (K-1) \cdot \frac{B^2 + C^2}{B^2 - C^2 + 2BT}$$

$$= 1 + (K-1) \cdot \frac{1}{1 + 2T\frac{B}{d^2} - 2\frac{C^2}{d^2}} \geq 1 + (K-1) \cdot \frac{1}{1 + 2T\frac{d}{d^2}}$$

$$= 1 + (K-1) \cdot \frac{1}{1 + 2T/d} > 1$$

Note that $K$ is a number larger than 1, we have shown that the FNA relationship in the theorem. Next, we calculate the chance that $f_2$ has a worse error than $f_2^*$. Based on the previous calculation, the FNR of any model is an increasing function w.r.t to the $x_1$ coordinate. Moreover, the model $f_2$'s center $\theta_2$ is uniformly distributed, under the arch $||\theta_2|| = K \cdot d$. The possibility of $f_2$ is worse than $f_2^*$ lies on the arch between $x_1 = KC$ and $X_1 = C$. Therefore, we find the probability of $\theta_2$ lying in this arch:

$$1 - \frac{\arccos \frac{C}{d}}{\arccos \frac{C}{Kd}} = 1 - \left(\frac{\pi}{2} - \frac{C}{d} + \Omega\left(\frac{C}{d}\right)^3\right) / \left(\frac{\pi}{2} - \frac{C}{Kd} + \Omega\left(\frac{C}{Kd}\right)^3\right)$$

$$\geq 1 - \left(\frac{\pi}{2} - \frac{C}{d} + \Omega\left(\frac{C}{d}\right)^3\right) / \left(\frac{\pi}{2} - \frac{C}{Kd}\right)$$

$\square$

# D OTHER DETECTION METHODS

In this section, we discuss other methodologies which can be used for ChatGPT text detection.

## D.1 SCORE-BASED METHODS

The work of Gehrmann et al. (2019) proposes the **GLTR** method. It records the rank (based on the probability score of an accessible model such as GPT2) of each token (in the vocabulary) and group them to 4 categories, which are top 10, top 100, and top 1,000 and others. Then, a linear model is trained using these 4 numbers as features for prediction.

**GPTZero** GPTZero.com is a public available tool designed for detecting LLM-generated texts by employing two principal linguistic metrics, which are "perplexity" and "burstiness". Specifically, **perplexity** is a measurement of how easy or difficult to understand and predict the next words in a sentence. A sentence with a lower perplexity typically flows smoothly and naturally, and allows humans to anticipate what might come next. Instead, sentences with higher perplexity are often regarded as confusing, difficult to follow, or unnatural in their structures and meanings. GPTZero

estimates perplexity through the output score from a fine-tuned[5] GPT-2 model. In detail, given a passage $x = (w_1, .., w_k)$, it calculates the perplexity score as:

$$\text{Perplexity} \propto \sum_{i=1}^{k} \log p(w_i | w_1, w_{i-1}),$$

where $p(\cdot)$ is the output probability of GPT-2. Notably, to test if a passage is generated from other LLMs like ChatGPT, the perplexity is also calculated by the same GPT-2 model. Beyond the perplexity, a high burstiness sentence refers to a sentence that exhibits a sudden, unexpected change or deviation from the typical language patterns or topic. GPTZero incorporates a "burstiness" check to analyze the text style as the content generated by LLMs tends to maintain consistency throughout the full passage. GPTZero calculates burstiness by using the standard deviation of the perplexity scores of each sentence in a given passage. GPTZero predicts a sentence or passage as LLM generated if the passage has a high perplexity or low burstiness.

### D.2 MODEL-BASED METHODS.

The model-based methods train deep learning models, which directly take the test passage as an input of the classification model. The GPT-2 Detector (Solaiman et al., 2019) fine-tunes a pre-trained language model RoBERTa Liu et al. (2019) models (RoBERTa-base and RoBERTa-large) for a binary classification task to distinguish GPT2-output data samples and human written samples from OpenWebText dataset. With the similar idea, *GPT-Sentinel* (Chen et al., 2023) and the work (Guo et al., 2023) also fine-tune existing text classification models, such as RoBERTa models, to detect ChatGPT texts. In detail, GPT-Sentinel Chen et al. (2023) trains a RoBERTa-base or a T5 model on a binary classification dataset, with human texts from OpenWebText and LLM generated texts obtained by using ChatGPT to rephrase texts from OpenWebText. Beyond RoBERTa, GPT-Sentinel also proposes a similar classification strategy based on another text model structure, called T5 model. The work (Guo et al., 2023) trains a RoBERTa classification model on HC-3 dataset, which includes human answers and ChatGPT answers to questions from sources such as Wikipedia and Reddit.

### D.3 SIMILARITY BASED METHODS

**GPT Paternity Test (GPT-Pat) (Yu et al., 2023)** proposes a different detection strategy beyond binary classification tasks. In particular, it assumes that if an LLM like ChatGPT is asked a same question for twice to generate two answers, these two answers tend to have a high similarity. Based on this assumption, given a test passage $x$, GPT-Pat first queries the ChatGPT model to generate a question based on the content of $x$, and inputs the question to ChatGPT again to query another answer $x'$. Then, it trains a similarity model (fine-tuned from RoBERTa) to measure the similarity of $x$ and $x'$. If $x$ and $x'$ are highly similar, it predicts $x$ as LLM generated. Similarly, **DNA-GPT** (Yang et al., 2023) also propose a strategy to let ChatGPT re-generate texts for comparison. However, the method in (Yang et al., 2023) does not involve the training process.

## E ABLATION STUDY ON OTHER LLMS

In this part, we provide additional empirical results to validate the conclusions in our paper for other LLMs, including GPT-4, Llama-2 (7b-chat) and PaLM2. In this part, we majorly repeat the experiment in Section 4.1 and Section 5 where we draw our main conclusions.

### E.1 GENERALIZATION TO UNSEEN PROMPTS

Recall Section 4.1, we claim: training with prompts with higher HC-Alignment can have better generalization to unseen prompts. Thus, in Figure 25, we repeat the similar experiment to study the generalization across prompts similar to Figure 1 (only focus on QA). In Figure 26, we calculate the HC-Alignment of different prompts generated by different models. In Figure 27, we also get a similar finding which is the prompts with higher HC-Alignment can have better generalization. Notably, an interesting finding is that the HC-Alignment of different prompts are not same in different LLMs.

---

[5]https://gptzero.me/technology

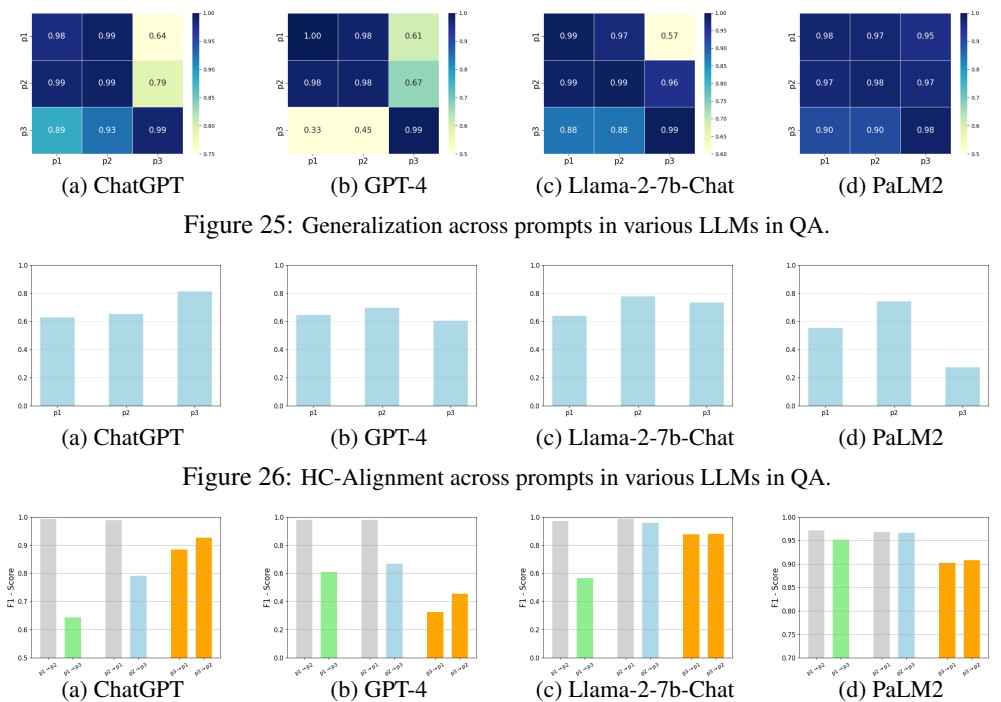

Figure 25: Generalization across prompts in various LLMs in QA.

Figure 26: HC-Alignment across prompts in various LLMs in QA.

Figure 27: HC-Alignment vs. Generalization across prompts in various LLMs in QA.

For example, in the task QA, the prompt P3 has highest HC-Alignment in ChatGPT. However, in other models like Llama-2 and PaLM2, the prompt P2 has the highest HC-Alignment.

## E.2 GENERALIZATION TO UNSEE TOPICS / TASKS

We conduct a similar experiment under Llama-2 generated texts to study the detection model's generalization to unseen topics. The table below reports the result for Llama-2 where the experimental setup resembles Table 7 in our paper. From our result in Llama-2, we can draw similar conclusions: the model can have a compromised performance when test on samples from unseen topics, which is denoted as "no transfer" in table below. However, if we apply transfer learning, such as fine-tuning (FT) or linear probing (LP) with a few samples, we can significantly improve the detection performance in the target domain.

Table 8: Transfer Learning (Topic-level) Performance via Linear Probing and Fine-tuning in QA under Llama2

| Target | finance | | | history | | | medical | | | science | | |
|---|---|---|---|---|---|---|---|---|---|---|---|---|
| | h→f | m→f | s→f | f→h | m→h | s→h | f→m | h→m | s→m | f→s | h→s | m→s |
| No Transfer | 0.979 | 0.841 | 0.980 | 0.977 | 0.902 | 0.994 | 0.762 | 0.885 | 0.796 | 0.975 | 0.991 | 0.881 |
| LP-5 | 0.975 | 0.898 | 0.975 | 0.977 | 0.926 | 0.995 | 0.923 | 0.955 | 0.965 | 0.979 | 0.990 | 0.910 |
| FT-5 | 0.952 | 0.921 | 0.911 | 0.974 | 0.942 | 0.958 | 0.873 | 0.956 | 0.887 | 0.952 | 0.962 | 0.930 |
| LP-Scratch-5 | 0.725 ± 0.028 | | | 0.702 ± 0.020 | | | 0.792 ± 0.031 | | | 0.751 ± 0.052 | | |
| FT-Scratch-5 | 0.821 ± 0.044 | | | 0.740 ± 0.067 | | | 0.734 ± 0.041 | | | 0.689 ± 0.087 | | |
| LP-10 | 0.980 | 0.912 | 0.974 | 0.979 | 0.932 | 0.995 | 0.936 | 0.956 | 0.970 | 0.986 | 0.986 | 0.912 |
| FT-10 | 0.961 | 0.942 | 0.974 | 0.975 | 0.966 | 0.988 | 0.903 | 0.970 | 0.890 | 0.960 | 0.980 | 0.945 |
| LP-Scratch-10 | 0.820 ± 0.062 | | | 0.794 ± 0.015 | | | 0.789 ± 0.050 | | | 0.818 ± 0.021 | | |
| FT-Scratch-10 | 0.903 ± 0.055 | | | 0.842 ± 0.043 | | | 0.865 ± 0.047 | | | 0.838 ± 0.046 | | |

