# OpenReview forum: "On the Generalization of Training-based ChatGPT Detection Methods"
_ICLR.cc/2024/Conference — Submitted to ICLR 2024_

### Official Review · Reviewer_vcVF · 2023-10-20

**Soundness:** 2 fair
**Presentation:** 3 good
**Contribution:** 2 fair
**Rating:** 6
**Confidence:** 2

**Summary:**

This paper presents an empirical evaluation on existing training-based "ChatGPT detection" methods. It has created a dataset which includes generated content prompted from ChatGPT and human written content. Analysis includes in-distribution evaluation, as well as OOD evaluation involving length-shift, topic- and domain-shift. Experiments show limited generalizability of training-based detection methods, and tend to overfit simplistic features. Meanwhile, it also suggests that these methods help with extracting transferable features that help generalize across domains.

**Strengths:**

This work studies an important and timely problem for detecting LLM-generated content. Experiments have conducted for in-distributed settings as well as OOD settings involving content length shift and topic/domain shift.
The feature attribution analysis is an interesting and novel angle of study in the context of LLM-generated content detection.

**Weaknesses:**

The study seeks for detecting content generated by ChatGPT, which is just an interface where the backend model keeps evolving. Hence, it is hard to say if the experimental results and analysis are reproducible and sustainable. In my opinion, this type of study should be conducted for a static LLM.
Length shift and domain/topic shift represent limited types of distribution shift that is easily detectable. The authors could have considered more implicit shift where content is paraphrased with syntax-controlled paraphrasing or style transfer, like those used in recent approaches for data pollution attack / defense.

Typos:

3.1:
don't -> do not

**Questions:**

It is common for training-based methods to have more limited generalizability since they somewhat overfit a specific data distribution. From another perspective, would any unsupervised OOD detection methods [1,2] apply to detecting LLM-generated content?

[1] Zhou, et al. Contrastive Out-of-Distribution Detection for Pretrained Transformers. EMNLP 2021
[2] Xu, et al. Contrastive Novelty-Augmented Learning: Anticipating Outliers with Large Language Models. ACL 2023

---

> ### Author Response · Authors · 2023-11-16
> **Response to Reviewer vcVF (Part 1)**
>
> We thank the reviewer for raising the concerns and questions. In the rebuttal, we will discuss: (1) Whether the experimental results and analysis are reproducible and sustainable? (2) How is the detection under more implicit distribution shift? (3) Would unsupervised OOD detection methods can be applied to detect LLM generated contents?
>
> **1. Whether the findings are reproducible and sustainable?**
>
> In our paper, the experimental results are based on the data collected from GPT-3.5 (turbo). Since the current ChatGPT is based on GPT-3.5, we claim that our paper is focused on ChatGPT.
>
> In the rebuttal and our revised paper, we follow the reviewer's suggestion to validate our major conclusions in other LLMs beyond GPT-3.5-turbo, including (1) other GPT models such as GPT-4; (2) other LLMs such as PaLM2 and Llama-2. We include the detailed results in Appendix E.1 in our revised submission. Overall, we can draw observations consistent to our conclusions in the paper. Below, we are happy to discuss some of these experimental results.
> 1. **Generalization to unseen prompts.** In our paper, we claim: training with prompts with higher HC-Alignment can have better generalization to unseen prompts. In Figure 26 in Appendix E.1, we calculate the HC-Alignment of different prompts generated by different models. In Figure 27, we also get a similar finding which is the prompts with higher HC-Alignment can have better generalization. Notably, an interesting finding is that the HC-Alignment of different prompts are not the same in different LLMs. For example, in the task QA, the prompt P3 has highest HC-Alignment in ChatGPT. However, in other models like GPT-4, Llama-2 and PaLM2, the prompt P2 has the highest HC-Alignment.
> 2. **Generalization to unseen topics / domain.** We conduct a similar experiment especially under Llama-2 generated texts to study the detection model's generalization to unseen topics in QA. The table below reports the result for Llama-2 where the experimental setup resembles Table 7 in our paper. From our result in Llama-2, we can draw similar conclusions: the model can have a compromised performance when testing on samples from unseen topics, which is denoted as "no transfer'' in table below. However, if we apply transfer learning, such as fine-tuning (FT) or linear probing (LP) with a few samples, we can significantly improve the detection performance in the target domain.
>
>
> |              |   |       | finance |       |   |   |   |       | history |       |   |   |   |       | medical |       |   |   |   |       | science |       |
> |--------------|---|-------|---------|-------|---|---|---|-------|---------|-------|---|---|---|-------|---------|-------|---|---|---|-------|---------|-------|
> |              |   | h->f  | m->f    | s->f  |   |   |   | f->h  | m->h    | s->h  |   |   |   | f->m  | h->m    | s->m  |   |   |   | f->s  | h->s    | m->s  |
> | No Transfer  |   | 0.979 | 0.841   | 0.980 |   |   |   | 0.977 | 0.902   | 0.994 |   |   |   | 0.762 | 0.885   | 0.796 |   |   |   | 0.975 | 0.991   | 0.881 |
> | LP-5         |   | 0.975 | 0.898   | 0.975 |   |   |   | 0.977 | 0.926   | 0.995 |   |   |   | 0.923 | 0.955   | 0.965 |   |   |   | 0.979 | 0.990   | 0.910 |
> | FT-5         |   | 0.952 | 0.921   | 0.911 |   |   |   | 0.974 | 0.942   | 0.958 |   |   |   | 0.873 | 0.956   | 0.887 |   |   |   | 0.952 | 0.962   | 0.930 |
> | LP-Scratch-5 |   | 0.725 | 0.725   | 0.725 |   |   |   | 0.702 | 0.702   | 0.702 |   |   |   | 0.792 | 0.792   | 0.792 |   |   |   | 0.751 | 0.751   | 0.751 |
> | FT-Scratch-5 |   | 0.821 | 0.821   | 0.821 |   |   |   | 0.740 | 0.740   | 0.740 |   |   |   | 0.734 | 0.734   | 0.734 |   |   |   | 0.689 | 0.689   | 0.689 |

---

> ### Author Response · Authors · 2023-11-16
> **Response to Reviewer vcVF (Part 2)**
>
> **(2) How is the detection under more implicit distribution shift?**
>
> In our paper, we consider the distribution shift due to the change of "prompts'', which is one type of implicit distribution shift. See our discussion in Section 3.1 and Appendix A.1, we build our dataset to include various prompts to obtain ChatGPT texts. For example, to let ChatGPT to write an essay, we ask ChatGPT to write in different styles. To let ChatGPT answer a question, we include the prompts to obtain formal answers and conversational answers. Based on this design, in Section 4.1, we consider the "prompt shift'', which is the scenario that the test samples are generated from prompts which are not included in training. Since the texts from different prompts are solving the same language task, this type of distribution shift is more implicit compared to topic or length shift. In our study in Section 4.1, we draw the conclusion: to train a detection model, if the ChatGPT texts obtained from certain prompts have a distribution closer to the human texts (a higher HC-Alignment as defined in Section 4.1), the detection model tends to have better generalization to unseen prompts.
>
>
> In our rebuttal, we are also happy to follow the reviewer's suggestion to consider the distribution shift caused by paraphrasing. We conduct an additional experiment to test the detection models when the ChatGPT texts are paraphrased by another LLM (PaLM2). In the table below, we repeat the basic experimental setting as Figure 1 (d), and we train and test the detection models under different prompts. Moreover, we use "Before'' and "After'' to denote the detection performance before and after the paraphrasing manipulation. Besides, we also report the corresponding performance drop after paraphrasing. From the result, we can see that all the detectors encounter the performance drop after paraphrasing. However, the detection model trained on prompt P3 is relatively robust, compared to P1 and P2. This is also consistent with our previous conclusion above, as P3 in QA has the highest HC-Alignment than P1 and P2 (see Figure 4 (a)).
>
> | Before |  P1  |  P2  |  P3  |   |   |   | After |      P1      |      P2      |      P3      |
> |:------:|:----:|:----:|:----:|:-:|:-:|:-:|:-----:|:------------:|:------------:|:------------:|
> |   P1   | 0.98 | 0.99 | 0.64 |   |   |   |   P1  | 0.91 (-0.07) | 0.88 (-0.11) | 0.60 (-0.04) |
> |   P2   | 0.99 | 0.99 | 0.79 |   |   |   |   P2  | 0.87 (-0.12) | 0.92 (-0.07) | 0.74 (-0.05) |
> |   P3   | 0.89 | 0.93 | 0.99 |   |   |   |   P3  | 0.86 (-0.03) | 0.91 (-0.02) | 0.95 (-0.04) |
>
> **(3) Would unsupervised OOD detection methods can be applied to detect LLM generated contents?**
>
> We agree with the reviewer that training-based methods tend to have limited generalizability. Thus, in Table 3 of our paper, we also investigate the existing unsupervised methods for ChatGPT detection. In Table 3, GPTZero [1] and DNA-GPT [2] are the methods which do not involve any training process. Specifically, GPTZero uses the perplexity and burtiness of a given text to determine whether it is generated by LLMs or written by human. DNA-GPT truncates a given text, and inputs the truncated text to ChatGPT to re-generate a text. Then, they compare the similarity between the given text and re-generated text to make prediction. In the rebuttal, we also include the Contrastive OOD detection method [3] mentioned by the reviewer. We treat human texts as in-distribution data and examine the detector's ability to identify LLM texts as OOD data. In the table below, we report the detection performance (F1-score) of this method. As a comparison, we also include the result of GPTZero, DNA-GPT and RoBERTa (a training-based method) from Table 3.
>
> |             | News | Review | Writing | QA   |
> |-------------|------|--------|---------|------|
> | GPTZero     | 0.94 | 0.90   | 0.89    | 0.90 |
> | DNA-GPT     | 0.90 | 0.90   | 0.92    | 0.82 |
> | Contrastive | 0.90 | 0.88   | 0.85    | 0.83 |
> | RoBERTa     | 1.00 | 0.99   | 1.00    | 0.98 |
>
>
> In this table for the training-based method, we report the detection performance when the training and test data are sampled from the same distribution. Under this setting, we can see the training-based method out-performs unsupervised methods. This is the reason our paper specifically focuses on training-based methods.
>
>
>
> **Reference**
>
> [1] GPTZero (GPTZero.com)
>
> [2] DNA-GPT: Divergent N-Gram Analysis for Training-Free Detection of GPT-Generated Text, Yang et al, 2023
>
> [3] Contrastive Out-of-Distribution Detection for Pretrained Transformers. Zhou, et al.

---

> ### Author Response · Authors · 2023-11-19
> **A friendly reminder**
>
> We are grateful for the useful comments provided by you. We hope that our answers have addressed your concerns. If you have any further concerns, please let us know. We are looking forward to hearing from you.

---

> ### Author Response · Authors · 2023-11-22
>
> We appreciate your reviews. We hope that our responses have adequately addressed your concerns. As the deadline for open discussion nears, we kindly remind you to share any additional feedback you may have. We are keen to engage in further discussion.

---

### Official Review · Reviewer_uBbK · 2023-11-01

**Soundness:** 2 fair
**Presentation:** 3 good
**Contribution:** 2 fair
**Rating:** 5
**Confidence:** 4

**Summary:**

The paper proposes a solution for detecting texts generated by ChatGPT.

Specifically, the authors focus on understanding the generalization behaviors of training-based detection methods under distribution shifts caused by various factors including prompts, text lengths, topics, and language tasks.

They also collect a new data set and present findings to guide future methodologies and data collection strategies for ChatGPT detection.

**Strengths:**

* The authors present a novel data set.
* The analysis is detailed and comprehensive.
* They provide insights on the data collection and domain adaption strategy.

**Weaknesses:**

* ChatGPT Direction seems to be not well motivated. It needs a why, not just a  what and a how.
* This work exclusively discusses the train-based methods, which are smaller in scope.

**Questions:**

Could you elaborate on ChatGPT Direction? How is it motivated?

---

> ### Author Response · Authors · 2023-11-16
> **Response to Reviewer uBbK**
>
> We thank the reviewer for the valuable comments and suggestions. In the rebuttal, we will provide discussions on: (1) what is "ChatGPT Direction'' and how is it motivated? and (2) why we focus on the training-based methods?
>
> **1. What is "ChatGPT Direction'' and how is it motivated?**
>
> To have a further explanation on "ChatGPT Direction'', it would be helpful for us to first recall the concept about "Irrelevant Direction''.  In Section 4 of our paper, we provide evidence showing that the detection models could learn "**irrelevant features**'' for prediction. For example, in Section 4.2, we show the possibility of a model to use "**length**'' as a deterministic feature to make prediction. As an evidence, Figure 5 (c) shows a detection model could have a high risk to predict short ChatGPT texts to be human written, because its training dataset has more short human written texts. However, since ChatGPT can be utilized to generate longer or shorter texts, the lengths should not be treated as the principal features for prediction. In our paper, we define the "**irrelevant direction**'' which contains these irrelevant features. Our analysis in Section 4.1 also finds that the collected ChatGPT generated texts from certain prompts could possibly differ from human texts in this "irrelevant direction'', which cause them have a poor HC-Alignment. In this way, the trained detection model can make predictions based on irrelevant features and have a poor generalization.
>
>
> Orthogonal to the irrelevant direction, we define the  "**ChatGPT direction**'' which is the direction that contains the principal features to distinguish human and ChatGPT texts. From our experiments, we demonstrate if one can prevent the detection model to learn irrelevant features, the models can have a better generalization, which validate the existence of principal features. Specifically: In Figure 5(b) in Section 4.2, if we train on ChatGPT texts with similar length distribution as human texts, the model can correctly predict both short and long texts. Based on these empirical findings, we verify the model can also learn principal features for prediction with better generalization. Thus, we are motivated to define this "ChatGPT Direction'' which contains these principal features. In our theoretical analysis in Section 4.3, the ChatGPT direction is the direction that determines the ground truth label (ChatGPT generated or human written) of data samples, and it is orthogonal to the irrelevant direction.
>
>
>
> **2. Why we focus on the training-based detection methods?**
>
>
> We are happy to discuss why we believe it is of great importance to have a specified and comprehensive study for training-based detection methods. First, the training-based methods have high "**in-distribution**'' performance. For example, in our empirical study in Table 3 in the paper, we compare the training-based methods with other methods beyond training-based methods, including "similarity-based'' methods and "score-based'' methods. From the result, we found that training-based methods can provide much more promising detection performance compared to other methods. Second, the training-based methods have high "**applicability**'' for ChatGPT detection and LLM-generated text detection in general. Notably, some well-known LLM-generated detection methods [1-2] are not applicable to black-box LLMs like ChatGPT. In specifics, the method [1] relies on the knowledge of model prediction score, and the method [2] needs to manipulate the text sampling process. However, the training-based methods only need to collect the human texts and LLM generated texts. This process is more general and applicable to all LLMs. Given the superior performance and wide applicability, we believe it is worth a comprehensive study especially for training based methods. Notably, in addition to the analysis, our collected dataset, HC-Var, is one of the most comprehensive and versatile dataset for the research of ChatGPT detection until now, which can also support the evaluation for various kind of detection techniques.
>
>
> **Reference**
>
> [1] DetectGPT: Zero-Shot Machine-Generated Text Detection using Probability Curvature, Mitchell et al, ICML 2023
>
> [2] A Watermark for Large Language Models, Kirchenbauer et al, ICML 2023

---

> ### Author Response · Authors · 2023-11-19
> **A friendly reminder**
>
> We are grateful for the useful comments provided by you. We hope that our answers have addressed your concerns. If you have any further concerns, please let us know. We are looking forward to hearing from you.

---

> ### Author Response · Authors · 2023-11-22
>
> We appreciate your reviews. We hope that our responses have adequately addressed your concerns. As the deadline for open discussion nears, we kindly remind you to share any additional feedback you may have. We are keen to engage in further discussion.

---

### Official Review · Reviewer_FWZZ · 2023-11-04

**Soundness:** 3 good
**Presentation:** 4 excellent
**Contribution:** 2 fair
**Rating:** 5
**Confidence:** 4

**Summary:**

This paper investigates the challenges in distinguishing between human-written and ChatGPT-generated texts. The paper makes significant contributions to the field by offering a detailed analysis of ChatGPT detection methods under various distribution shifts. But I think most of findings seem obvious and it is more technical than scientific.

**Strengths:**

- **Comprehensive Investigation**: The paper conducts a thorough analysis of the generalization behaviors of existing methods under distribution shifts caused by various factors like prompts, text lengths, topics, and language tasks.

- **New Dataset**: The authors contribute to the field by collecting a new dataset containing both human and ChatGPT-generated texts, facilitating in-depth studies on the detection methods.

- **Insightful Findings**: The research uncovers insightful findings, providing valuable guidance for the development of future methodologies and data collection strategies for ChatGPT detection.

**Weaknesses:**

- The authors used three prompts in Figure 1; but in general, users might use various prompts. This is not aligned with real user usage,
- The experiments are limited to CHATGPT, we do not know whether these conclusions still hold in GPT4 or other open-source LLMs.
- Most findinds seem obvious.

**Questions:**

- Can we have a setting to mix various prompts, for example in Figure 1. This could be aligned with the real user usage
- How are these findings valid to other LLMs?

---

> ### Author Response · Authors · 2023-11-16
> **Response to Reviewer FWZZ (Part 1)**
>
> We thank the reviewer for raising valuable comments and questions. In our rebuttal, we will discuss: (1) How is the detection when training with multiple prompts? (2) whether the findings are valid for other LLMs? and (3) why findings in our paper are significant?
>
> **1. How is the detection when training with multiple prompts?**
>
> We follow the reviewer's suggestion to conduct an experiment similar to the study in Figure 1, but training with multiple prompts. In detail, for the language tasks "Review'' and "QA'', we consider to train the detection models with ChatGPT texts from 2 prompts, and we test the trained model on each of the 3 prompts. The table below reports the F-1 score of the detection models.
>
> | Review | P1   | P2   | P3   |   |   |   |   | QA    | P1   | P2   | P3   |
> |--------|------|------|------|---|---|---|---|-------|------|------|------|
> | P1&P2  | 1.00 | 1.00 | 0.94 |   |   |   |   | P1&P2 | 1.00 | 1.00 | 0.84 |
> | P2&P3  | 1.00 | 1.00 | 0.99 |   |   |   |   | P2&P3 | 1.00 | 1.00 | 0.98 |
> | P1&P3  | 1.00 | 1.00 | 1.00 |   |   |   |   | P1&P3 | 1.00 | 0.99 | 0.99 |
>
> From the results, we can have the following observations: (1) if  a prompt during test appears in the training dataset, the model has a high detection performance on this prompt. (2) if a prompt during test does not appear in the training dataset, but the training set contains a prompt with a high generalization to this prompt (based on the result from Figure 1), the model can also have a high detection performance on this prompt. (3) If we train the model using ChatGPT texts from  "P1 \& P2", the model has a relatively lower detection generalization to P3. This observation is consistent to our major claim in the analysis in Section 4.1. Recall Section 4.1, we claim: for models trained on prompts with lower HC-Alignment, they may have a poorer generalization. Notably, in "Review'' and "QA'', both P1 and P2 has a lower HC-Alignment than P3 (see Figure 4 (a)). Thus, training on them together might also be insufficient to correctly detect texts from P3. Thus, we see a lower detection performance in the table.
>
> In practice,  multiple prompts can be utilized to train the detection model. However, it is still possible that the test data contains texts from new prompts. The analysis in this table also supports our suggestion in Section 4.1 to avoid only collecting samples far away from human data. To have a clear presentation for this point, we only show the scenario when training with one prompt in the main paper. We have followed the reviewer's suggestion to add this analysis into the appendix of our revised paper.

---

> ### Author Response · Authors · 2023-11-16
> **Response to Reviewer FWZZ (Part 2)**
>
> **2. Whether the findings are valid for other LLMs?**
>
> We follow the reviewer's suggestion to validate our major conclusions in other LLMs beyond ChatGPT, including GPT-4, Llama-2 (7b-chat) and PaLM2. We include the detailed results in Appendix E.1 in our revised submission. Overall, we can draw observations consistent to our conclusions in the paper. Below, we are happy to discuss some of these experimental results.
> 1. **Generalization to unseen prompts.** In our paper, we claim: training with prompts with higher HC-Alignment can have better generalization to unseen prompts. In Figure 26 in Appendix E.1, we calculate the HC-Alignment of different prompts generated by different models. In Figure 27, we also get a similar finding which is the prompts with higher HC-Alignment can have better generalization. Notably, an interesting finding is that the HC-Alignment of different prompts are not the same in different LLMs. For example, in the task QA, the prompt P3 has highest HC-Alignment in ChatGPT. However, in other models like Llama-2 and PaLM2, the prompt P2 has the highest HC-Alignment.
>
> 2. **Generalization to unseen topics / domain.** We conduct a similar experiment especially under Llama-2 generated texts to study the detection model's generalization to unseen topics in QA. The table below reports the result for Llama-2 where the experimental setup resembles Table 7 in our paper. From our result in Llama-2, we can draw similar conclusions: the model can have a compromised performance when testing on samples from unseen topics, which is denoted as ``no transfer'' in table below. However, if we apply transfer learning, such as fine-tuning (FT) or linear probing (LP) with a few samples, we can significantly improve the detection performance in the target domain.
>
> |              |   |       | finance |       |   |   |   |       | history |       |   |   |   |       | medical |       |   |   |   |       | science |       |
> |--------------|---|-------|---------|-------|---|---|---|-------|---------|-------|---|---|---|-------|---------|-------|---|---|---|-------|---------|-------|
> |              |   | h->f  | m->f    | s->f  |   |   |   | f->h  | m->h    | s->h  |   |   |   | f->m  | h->m    | s->m  |   |   |   | f->s  | h->s    | m->s  |
> | No Transfer  |   | 0.979 | 0.841   | 0.980 |   |   |   | 0.977 | 0.902   | 0.994 |   |   |   | 0.762 | 0.885   | 0.796 |   |   |   | 0.975 | 0.991   | 0.881 |
> | LP-5         |   | 0.975 | 0.898   | 0.975 |   |   |   | 0.977 | 0.926   | 0.995 |   |   |   | 0.923 | 0.955   | 0.965 |   |   |   | 0.979 | 0.990   | 0.910 |
> | FT-5         |   | 0.952 | 0.921   | 0.911 |   |   |   | 0.974 | 0.942   | 0.958 |   |   |   | 0.873 | 0.956   | 0.887 |   |   |   | 0.952 | 0.962   | 0.930 |
> | LP-Scratch-5 |   | 0.725 | 0.725   | 0.725 |   |   |   | 0.702 | 0.702   | 0.702 |   |   |   | 0.792 | 0.792   | 0.792 |   |   |   | 0.751 | 0.751   | 0.751 |
> | FT-Scratch-5 |   | 0.821 | 0.821   | 0.821 |   |   |   | 0.740 | 0.740   | 0.740 |   |   |   | 0.734 | 0.734   | 0.734 |   |   |   | 0.689 | 0.689   | 0.689 |
>
> **3. Whether the finding is obvious?**
>
> The findings of this paper are established through investigation based on our collected dataset HC-Var. Thus, we would first highlight the contribution of our dataset. As mentioned in Section 3 in our paper, compared to other datasets for ChatGPT detection, our dataset is the first one to consider the factors including the prompts and lengths during data collection. Notably, before our dataset, one of the most investigated datasets for ChatGPT detection, HC-3 [1], only considers one prompt to inquire ChatGPT outputs and also neglects the impact of lengths. Therefore, our collected dataset is a good supplementary to the literature of existing works.
>
> Our findings can be drawn thanks to the diversity of different factors that are provided by our dataset.
> Therefore, we respectively disagree that our findings are obvious. In fact, our findings are novel compared to existing works. Especially, through our analysis on the generalization across different prompts, we disclose the pitfall of only collecting samples far away from human data. This is not adequately addressed by previous works. Moreover, each of our findings has been subjected to rigorous examination, including using numerical metrics such as MAUVE [2], visualization, and theories. These findings can also be treated as preliminary insights which pave the way for further research.
>
>  **Reference:**
>
> [1] How Close is ChatGPT to Human Experts? Comparison Corpus, Evaluation, and Detection, Guo et al, 2023.
>
> [2] MAUVE: Measuring the Gap Between Neural Text and Human Text using Divergence Frontiers, Pillutla, et al, 2021

---

> ### Author Response · Authors · 2023-11-19
> **A friendly reminder**
>
> We are grateful for the useful comments provided by you. We hope that our answers have addressed your concerns. If you have any further concerns, please let us know. We are looking forward to hearing from you.

---

> ### Author Response · Authors · 2023-11-22
>
> We appreciate your reviews. We hope that our responses have adequately addressed your concerns. As the deadline for open discussion nears, we kindly remind you to share any additional feedback you may have. We are keen to engage in further discussion.

---

### Author Response · Authors · 2023-11-22
**Close to end of the discussion period**

Dear Reviewers, PC, AC,

We wish to express our sincere gratitude for the assistance and support from all of you. All the time, we appreciate the spirit of ICLR of encouraging the discussion between reviewers and authors.

**However, we still did't receive any response to our paper**, although we tried our best to write detailed rebuttals to every reviewer several days ago. As it is close to the end of the discussion period, we would request **one opportunity** again for the discussion. We will stay online to welcome any questions and provide further clarifications.

---

### Author Response · Authors · 2023-11-23

Dear Reviewers,

We wish to express our sincere gratitude for the assistance and support from all of you. All the time, we appreciate the spirit of ICLR of encouraging the discussion between reviewers and authors.

However, we still did't receive any response to our paper, although we tried our best to write detailed rebuttals to every reviewer several days ago. As it is close to the end of the discussion period, we would request **one opportunity** again for the discussion for the last time. We will **stay online until the close of discussion period** to welcome any questions and provide further clarifications.

---

### Meta-Review · Area_Chair_QLoE · 2023-12-12

**Metareview:**

This paper presents a method of detecting whether a piece of text is generated by ChatGPT.  The main idea is to train supervised models for this task and the authors focus on three pretty broad prompt types.  They report empirical results on both in domain and out of domain prompts and have a thorough empirical setup.

Strengths:  Strong experimental setup, the overall motivation is good, new dataset, well written paper.

Weaknesses:  Overall, I find that this paper is a step in the right direction, but it is limited by on a small set of prompt types, when real users have a very broad swath of prompts that they may input to utilize an LLM.  In addition, most of the study focuses on ChatGPT, which is not an ideal LLM to probe since it changes all the time--a larger study on reproducible LLMs (at least for a significant time horizon like 2 years instead of ChatGPT which changes every few weeks) would have been a stronger design.  These are points brought up by the reviewers and while the reviewers did not engage a lot during the discussion, I did not find all the responses to cover enough ground to help accept the paper.

**Justification For Why Not Higher Score:**

Please see the discussion above--the paper could be made much stronger if it is rewritten with the suggestions from the metareview and from the reviews below.

**Justification For Why Not Lower Score:**

Not applicable.

---

### Decision · Program_Chairs · 2024-01-16

Reject